# OwLore: Outlier-weighed Layerwise Sampled Low-Rank Projection for LLM Fine-tuning

## Abstract

The rapid advancements in Large Language Models (LLMs) have revolutionized various natural language processing tasks. However, the substantial size of LLMs presents significant challenges in training or fine-tuning. While parameter-efficient approaches such as low-rank adaptation (LoRA) have gained popularity, they often compromise performance compared to full-rank fine-tuning. In this paper, we propose *Outlier-weighed Layerwise Sampled Low-Rank Projection (**OwLore**)*, a new memory-efficient fine-tuning approach, inspired by the layerwise outlier distribution of LLMs. Unlike LoRA, which adds extra adapters to all layers, OwLore strategically assigns higher sampling probabilities to layers with more outliers, selectively sampling only a few layers and fine-tuning their pre-trained weights. To further increase the number of fine-tuned layers without a proportional rise in memory costs, we incorporate gradient low-rank projection, further boosting the approach's performance. Our extensive experiments across various architectures, including LLaMa2, LLaMa3, and Mistral, demonstrate that OwLore consistently outperforms baseline approaches, including full fine-tuning. Specifically, it achieves up to a 1.1% average accuracy gain on the Commonsense Reasoning benchmark, a 3.0% improvement on MMLU, and a notable 10% boost on MT-Bench, while being more memory efficient. OwLore allows us to fine-tune LLaMa2-7B with only 21GB of memory. Our code is submitted.

## 1 Introduction

The rapid advancements in artificial intelligence (AI) driven by Large Language Models (LLMs) have fundamentally transformed how people work and communicate. The impressive language capabilities of LLMs enable a single model to handle various tasks simultaneously, including but not limited to natural language understanding (Brown et al., 2020; Touvron et al., 2023), text generation (Kocoń et al., 2023; Anil et al., 2023), machine translation (Jiao et al., 2023), and programming (Surameery & Shakor, 2023; Tian et al., 2023). However, the massive size of LLMs presents significant challenges for practical applications and deployment.

To address these challenges, various parameter-efficient approaches have been proposed, including prompt tuning (Lester et al., 2021; Liu et al., 2021a), adaptors (Houlsby et al., 2019; He et al., 2021), and low-rank adaptation (LoRA) (Hu et al., 2021; Dettmers et al., 2024). These approaches enable the fine-tuning of pre-trained LLMs with substantially fewer trainable parameters, making LLM fine-tuning more feasible in practice. Among these, LoRA (Hu et al., 2021) stands out for its re-parameterization technique of the pre-trained weight matrix $W \in \mathbb{R}^{m \times n}$, expressed as $W_0 + AB$, where $A \in \mathbb{R}^{m \times r}$, $B \in \mathbb{R}^{r \times n}$, and $r \ll \min(m, n)$. By fine-tuning only the low-rank adaptor $AB$ while keeping the pre-trained weight $W_0$ frozen, LoRA significantly reduces the memory usage and computational costs associated with fine-tuning LLMs, rapidly becoming the preferred method for such tasks. Despite its efficiency, recent research has highlighted the inferior performance of low-rank reparameterization compared to full-rank updates in both fine-tuning scenarios (Xia et al., 2024; Biderman et al., 2024) and pre-training contexts (Lialin et al., 2023b; Zhao et al., 2024). These findings underscore the need for further exploration into balancing training efficiency with model performance, particularly in the context of large-scale language models.

In a parallel vein, layerwise sampling for LLM fine-tuning appears to be a promising alternative for more effectively preserving the full fine-tuning trajectory. Pan et al. (2024) introduced LISA, a

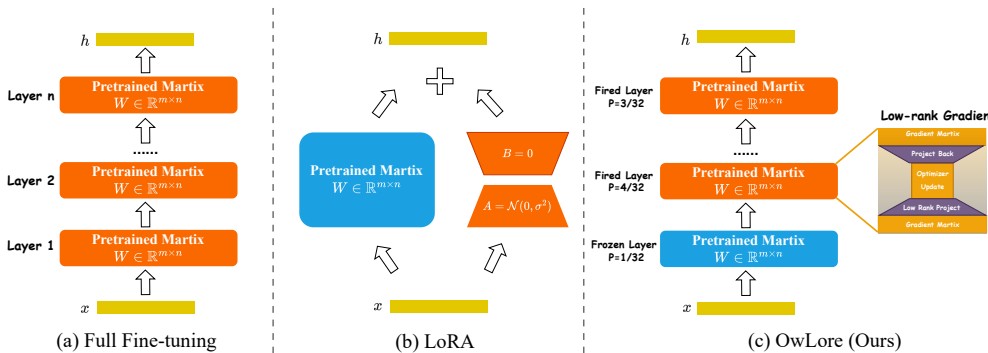

Figure 1: The comparison among Full Fine-tuning, training with LoRA, and Owlore. Blue modules are frozen, while orange modules are activated. OwLore non-uniformly samples layers to fine-tune models with low-rank gradients.

novel fine-tuning approach for LLMs that integrates the concept of importance sampling (Kloek & Van Dijk, 1978; Zhao & Zhang, 2015) into the fine-tuning process. Instead of using extra adaptors for all layers, LISA only samples a couple of layers and directly fine-tunes their pre-trained weights, demonstrating compelling performance gain over LoRA. For simplicity, we refer to approaches that fine-tune by sampling layers as *sampling-based fine-tuning* throughout this paper.

However, it remains a challenge to find an optimal layerwise sampling method for pre-trained LLMs. For instance, our preliminary investigation reveals the following intriguing observations: ❶ the sampling strategy used by LISA is sub-optimal, failing to compete with a very simple baseline, i.e. monotonic decreasing sampling probabilities from top to bottom layers as shown in Table 1; ❷ The sampled layers are fine-tuned in a full-rank manner, which means that increasing the number of unfrozen layers will significantly raise the memory overhead, as shown in Table 2. As noted in Pan et al. (2024), LISA's performance improves with higher rank levels. Therefore, full-rank training constrains the potential performance gains of LISA. Although memory-efficient low-rank training methods like GaLore (Zhao et al., 2024) have shown promising results in pre-training, it performs no better than LoRA in the scenario of fine-tuning (Zhao et al., 2024). These observations motivate further exploration into more principled methodologies for sample-based fine-tuning, aiming to enhance both performance and memory efficiency.

**Overview.** In this paper, we introduce Outlier-weighted Layerwise Sampled Low-Rank Projection (**OwLore**), a novel memory-efficient method for fine-tuning large language models (LLMs). Our approach leverages the unique characteristic of LLMs where certain features and weights—referred to as outliers—have significantly larger magnitudes than the rest (Kovaleva et al., 2021; Puccetti et al., 2022; Dettmers et al., 2022; Yin et al., 2024). Based on the principle that layers with more outliers are more critical for fine-tuning, we assign higher sampling probabilities to layers with a greater concentration of outliers, essentially forming a rich-get-richer phenomenon, substantially improving the fine-tuning performance. Our results verify that our outlier-weighted layerwise importance score outperforms previous layerwise importance scores such as Relative Magnitude (Samragh et al., 2023) and Block Influence (Men et al., 2024). To further increase the number of fine-tuned layers without a proportional rise in memory costs, we incorporate gradient low-rank projection (Zhao et al., 2024), which further provides a performance boost to our approach. The combination of sampling-based fine-tuning with gradient low-rank projection not only enhances the performance-memory trade-off of sampling-based fine-tuning but also boosts the effectiveness of gradient low-rank projection in fine-tuning.

Our extensive experiments across various architectures including LLaMa2 (Touvron et al., 2023), LLaMa3 (Meta, 2024), and Mistral (Jiang et al., 2023) demonstrate that OwLore consistently outperforms its baseline approaches including full-parameter fine-tuning. OwLore achieves up to a 1.1% average accuracy gain on the Commonsense Reasoning benchmark, a 3.0% improvement on MMLU, and a notable 10% boost on MT-Bench, while being more memory efficient. Notably, OwLore allows fine-tuning LLaMa2-7B with only 21GB of memory. *Note that different from LoRA which adds additional adaptors, OwLore directly fine-tunes the original pre-trained weights, preserving the original optimization trajectory while being more memory-efficient.*

## 2 RELATED WORK

**Parameter-Effieient Fine-Tuning (PEFT).** PEFT is proposed to reduce the prohibitive cost of LLM fine-tuning. Various techniques have been proposed in this dynamic field. For instance, prompt tuning only optimizes input tokens or embeddings while keeping the rest of the model frozen, as demonstrated in studies (Lester et al., 2021; Li & Liang, 2021; Hambardzumyan et al., 2021; Zhong et al., 2021). Layer-freezing techniques (Liu et al., 2021b; Brock et al., 2017; Li et al., 2024) enhance training and fine-tuning efficiency by freezing parts of the layers. Adapter methods (Houlsby et al., 2019; He et al., 2021; Mahabadi et al., 2021; Diao et al., 2022), incorporate a small auxiliary module within the model's architecture, which becomes the exclusive focus of updates during training, thus minimizing the number of trainable parameters and optimizer states. Among these techniques, Low-Rank Adaptation (LoRA) (Hu et al., 2021) gains massive attention by applying low-rank matrices to approximate weight changes during fine-tuning, which can be merged into the pre-trained weights, leading to no inference overhead. LoRA has been enhanced through various modifications (Zhang et al., 2023; Renduchintala et al., 2023; Sheng et al., 2023; Liu et al., 2024; Kopiczko et al., 2023; Dettmers et al., 2024; Zhao et al., 2024) aimed at improving performance and efficiency. Recently, low-rank has also been explored to pre-train LLM from scratch (Lialin et al., 2023a; Zhao et al., 2024). GaLore (Zhao et al., 2024) projects the gradient into a low-rank subspace for the update to enable full-parameter learning while significantly reducing memory usage during optimization. BAdam (Luo et al., 2024) partitions the entire model into distinct blocks and utilizes a block coordinate descent framework to update each block individually, either in a deterministic or random sequence.

**Layerwise Sampling for LLM Fine-tuning.** Importance sampling is a powerful statistical technique used in machine learning to estimate properties of a particular distribution by sampling from a different, more convenient distribution. Recently, Pan et al. (2024) explored the idea of importance sampling to LLM fine-tuning, with the key idea of sampling only $\gamma$ layers at each step to fine-tuning while keeping the rest of layers frozen. The proposed method, Layerwise Importance Sampled AdamW (LISA), outperforms LoRA by a large margin on various benchmarks and even outperforms full parameters training under certain settings. Inspired by LISA, our paper advances the performance of layerwise sampling for LLM fine-tuning, by addressing a couple of shortfalls of LISA.

## 3 LIMITATIONS OF LAYERWISE IMPORTANCE SAMPLED ADAMW (LISA)

In this section, we first introduce LISA's algorithm and then present our findings of two key limitations of LISA: the shortcomings of its sampling approach and the significant memory overhead associated with the sampled layers.

**Layerwise Importance Sampled AdamW (LISA).** Pan et al. (2024) conducted an in-depth analysis of LoRA's training dynamics across layers and revealed an unusual skew in the distribution of layerwise weight norms, particularly towards the top layer and/or the bottom layer [1], where the norms are significantly larger compared to other layers. Building upon this insight, the authors proposed LISA, a novel fine-tuning approach for LLMs, which incorporates the concept of importance sampling (Kloek & Van Dijk, 1978; Zhao & Zhang, 2015) into the fine-tuning process. In LISA, layers of the base model are sampled to be unfrozen during training based on a prescribed probability, with the exception of the top and bottom layers, which remain activated throughout the process. Given a network with $N_L$ layers, the sampling probability of layer $\ell$ is given as follows:

$$p_\ell = \begin{cases} 1.0, & if\ \ell = 1\ or\ \ell = N_L, \\ \gamma/N_L & else. \end{cases} \quad (1)$$

where $\gamma$ controls the expected number of unfrozen layers during optimization. Since LISA does not require additional adaptors and only fine-tunes an expected $\gamma$ layers, it notably reduces the memory usage of LLM fine-tuning.

---

[1] Please note that in LISA, the terms 'top' and 'bottom' layers refer to the embedding layer and the LLM head layer, respectively, rather than the first and last Transformer blocks.

## 3.1 LIMITATIONS OF LISA

While demonstrating promising results, we observe that the LISA algorithm inherently has two shortcomings that constrain its memory-performance trade-off:

**i. The middle layers of LISA are sampled uniformly, which can result in suboptimal performance.** To verify this point, we conduct a small experiment where we replace the uniform sampling with a very simple baseline, i.e. monotonic decreasing sampling, where the sample probability is monotonically decreasing from shallow layers to deep layers (noted as LISA-D). Table 1 shows that this simple sampling method often outperforms uniform sampling, verifying our concern.

Table 1: Fine-tuning performance of LLaMA2-7B with various dataset.

| Model | Method | BoolQ | PIQA | SIQA | HellaSwag | WinoGrande | OBQA | Average |
|---|---|---|---|---|---|---|---|---|
| Llama2-7B | LISA | 82.0 | 79.9 | 33.5 | 59.7 | 79.6 | **38.8** | 62.25 |
| Llama2-7B | LISA-D | **85.1** | **79.9** | **33.8** | **59.8** | **79.7** | 38.4 | **62.78** |

**ii. The sampled layers of LISA are fine-tuned in a full-rank manner, causing a significant memory increase as the number of sampled layers increases.** To illustrate this, we fine-tune LLaMA2-7B on the GSM8K training set and report the GSM8K score and memory usage of LISA with various numbers of sampled layers $\gamma$, as shown in Table 2. The memory requirement of LISA rises significantly from 23GB to 36GB as $\gamma$ increases from 1 to 12. Similarly, the performance improves consistently with the increase in sampled layers. Since sampling more layers results in stronger fine-tuning performance, it is crucial to reduce the associated memory overhead as the number of sampled layers grows.

Table 2: GSM8K scores/memory usage for fine-tuning LLaMA2-7B with various expected sampled layers $\gamma$.

| Model | Method | $\gamma = 1$ | $\gamma = 2$ | $\gamma = 4$ | $\gamma = 8$ | $\gamma = 12$ |
|---|---|---|---|---|---|---|
| LLaMA2-7B | LISA | 16.8/23G | 18.8/25G | 19.8/27G | 19.9/32G | 21.7/36G |
| LLaMA2-7B | OwLore | 20.0/21G | 21.9/22G | 23.5/23G | 25.7/25G | 27.8/27G |

## 4 OUTLIER-WEIGHED LAYERWISE LOW-RANK PROJECTION (OWLORE)

In this section, we introduce our approach, Outlier-weighed Layerwise Low-Rank Projection (**OwLore**). We will discuss the underlying rationales, present preliminary results, and detail the algorithm design.

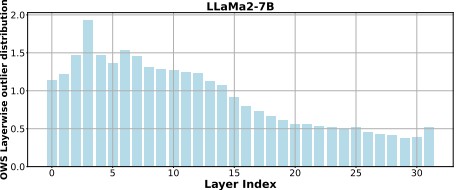 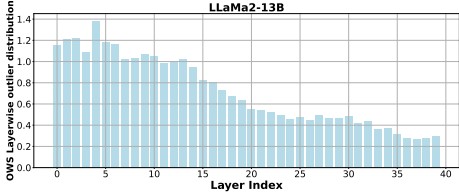

Figure 2: OWS Layerwise outlier distribution of LLaMa2 of Equation 2.

The above findings shed light on a principle for designing non-uniform layerwise sampling for LLM fine-tuning: layers with higher outlier ratios should be prioritized during the fine-tuning process. This forms the foundation of our proposed method, Outlier-weighed Layerwise Low-Rank Projection (OwLore), which we will present in detail.

**Outlier-Weighed Sampling (OWS).** Although LISA-D achieves good performance, it is more desirable to seek a more principled approach to determine the layerwise sampling probability. In the context of LLMs, we get inspiration from the unique characteristic of LLMs, outliers, defined as features and weights exhibiting significantly larger magnitudes compared to the majority of others (Kovaleva et al., 2021; Puccetti et al., 2022; Dettmers et al., 2022; Sun et al., 2023; Yin et al., 2024).

Our motivation stems from the crucial role outliers play in optimizing the performance of LLMs. We believe that layers containing more outliers are more important for fine-tuning. Therefore, we assign higher sampling probabilities to layers with more outliers during fine-tuning, leading to a substantial improvement in performance. To formulate, let us consider the input of a layer as $\mathbf{X}$ with dimensions $(N \times L, C_{\texttt{in}})$, where $N$ and $L$ represent the batch and sequence dimensions, respectively; and the weight matrix $\mathbf{W}$ has dimensions $(C_{\texttt{out}}, C_{\texttt{in}})$. Outlier score of weight $\mathbf{W}_{\texttt{ij}}$ is computed as $\mathbf{A}_{\texttt{ij}} = \|\mathbf{X}_{\texttt{j}}\|_2 \cdot |\mathbf{W}_{\texttt{ij}}|$. Here, $\|\mathbf{X}_{\texttt{j}}\|_2$ is the $\ell_2$ norm of input feature connected to the weight.

We first calculate the layerwise outlier distribution of a $N_L$-layer as $[D_1, D_2, ..., D_{N_L}]$, where $D_\ell$ characterizes the outlier ratio of layer $\ell$:

$$D_\ell = \frac{\sum_{i=1}^{C_{\texttt{out}}} \sum_{j=1}^{C_{\texttt{in}}} \mathbb{I}(\mathbf{A}_{\texttt{ij}}^\ell > \tau \cdot \bar{\mathbf{A}}^\ell)}{C_{\texttt{in}} C_{\texttt{out}}}, \tag{2}$$

where $\bar{\mathbf{A}}^\ell$ is the mean of $\mathbf{A}^\ell$ and $\mathbb{I}(\cdot)$ is the indicator function, returning 1 if $\mathbf{A}_{\texttt{ij}}^\ell$ is larger than $\tau \cdot \bar{\mathbf{A}}^\ell$, else 0. The layerwise outlier distribution essentially counts up weights whose outlier score is $\tau^2$ times greater than that layer's average outlier score. Larger $D$ means more outliers are presented in the corresponding layer. The sampling probability $p_\ell$ of layer $\ell$ is then calculated as $p_\ell = \gamma D_\ell / \sum_{i=1}^{N_L} D_i$, where $\gamma$ is the hyperparameter inherited from LISA to control the expected number of unfreeze layers during optimization. At each iteration, only the sampled layers will be fine-tuned, while the remaining layers are kept frozen. OWS naturally leads to a *rich-get-richer*[3] phenomenon, where layers containing more outliers during pre-training are sampled and fine-tuned more frequently. The visualization of layerwise outlier distribution of OWS is illustrated in Figure 2.

Table 3: Fine-tuning performance of LLaMA2-7B with various sampling approaches.

| Model | Sampling Method | BoolQ | PIQA | SIQA | HellaSwag | WinoGrande | OBQA | Average |
|---|---|---|---|---|---|---|---|---|
| LlaMa2-7B | Uniform (Pan et al., 2024) | 82.0 | 79.9 | 33.5 | 59.7 | 79.6 | 38.8 | 62.25 |
| LlaMa2-7B | BI (Men et al., 2024) | 82.8 | 79.6 | 33.2 | 60.3 | 80.4 | 36.6 | 62.15 |
| LlaMa2-7B | RM (Samragh et al., 2023) | 83.4 | 80.4 | 33.1 | 57.7 | 79.8 | 37.4 | 61.97 |
| LlaMa2-7B | OWS (ours) | 85.1 | 80.3 | 34.5 | 59.8 | 80.5 | 39.2 | **63.23** |

We compare OWS with other layerwise importance scores for sampling-based fine-tuning, including Uniform (Pan et al., 2024), Relative Magnitude (RM) (Samragh et al., 2023) and Block Influence (BI) (Men et al., 2024) in Table 3. OWS consistently performs better than other layer importance scores.

**Gradient Low-rank Projection.** Outlier-weighed sampling addresses our first research question: how to optimally sample layers for sampling-based LLM fine-tuning. To tackle the second issue of the substantial memory cost associated with an increasing number of unfrozen layers, we propose to integrate outlier-weighed sampling with gradient low-rank training. In this approach, the sampled layers are updated in a low-rank manner. Specifically, we adopt GaLore proposed in Zhao et al. (2024), wherein for each sampled layer, the gradient matrix is projected into a low-rank subspace using Singular Value Decomposition (SVD). The optimizer states are subsequently updated in the corresponding low-rank subspace with a rank level of $r$, significantly reducing the memory cost of optimization. We update the gradient subspace every 200 iterations to better capture the dynamic trajectory of fine-tuning. It is important to note that, while GaLore itself is not a novel approach, we are the first to demonstrate its effectiveness in the context of sampling-based fine-tuning. Combining sampling-based fine-tuning with gradient low-rank projection not only enhances the performance-memory trade-off of sampling-based fine-tuning but also boosts the effectiveness of gradient low-rank projection in LLM fine-tuning, which is beyond the scope of the original paper.

The above two innovations significantly boost the memory efficiency of OwLore, unlocking the performance-memory trade-off of sampling-based fine-tuning. At the macro level, we dynamically sample a limited number of layers to fine-tune at each iteration. At the micro level, each sampled layers are updated with low-rank gradients. Since the sampled layers are updated in the low-rank subspace, we can efficiently increase the number of sampled layers $\gamma$ with only a marginal increase in memory cost compared to LISA. Additionally, as we sample only a few layers at each fine-tuning

---

[2]We empirically find $\tau = 13$ consistently works well and choose it for all experiments in this paper.

[3]Here, the "rich-get-richer" phenomenon refers to layers with higher initial outlier scores being sampled more frequently for fine-tuning, which leads to these layers being better trained. However, this does not imply that these layers will accumulate more outliers over time as a result of the fine-tuning process.

---

**Algorithm 1:** Outlier-weighed Layerwise Low-Rank Projection (OwLore)

---

**Require:** number of layers $N_L$, number of training iterations $T$, sampling period $K$, sampled layers $\gamma$, rank level $r$, and $\mathcal{U}(0, 1)$ refers to a uniform sampling.

**for** $\ell \leftarrow 1$ **to** $N_L$ **do**
    Calculate outlier ratio $D_j$ using the Equation 2
    $p_\ell \leftarrow \frac{\gamma D_\ell}{\sum_{j=1}^{N_L} D_j}$         ▷ *Mapping layerwise outlier distribution to sampling probability.*

**for** $i \leftarrow 0$ **to** $T/K - 1$ **do**
    **for** $\ell \leftarrow 1$ **to** $N_L$ **do**
        **if** $\mathcal{U}(0, 1) > p_\ell$ **then**
            Freeze layer $\ell$

    **if** *Owlore-Full* **then**
        Run AdamW for $K$ iterations   ▷ *For Owlore-Full, we use the default AdamW optimizer with full ranks.*
    **if** *Owlore* **then**
        Run gradient low-rank update for $K$ iterations using GaLore Zhao et al. (2024) ▷ *For OwLore, we use GaLore Zhao et al. (2024) with low-rank gradients as shown in Algorithm A.*

---

iteration, we can increase the rank levels $r$ without significantly raising the memory requirements compared to LoRA. Memory usage analysis is given in Section 5.3. We perform a small search and find that $\gamma = 5$ and $r = 128$ consistently give us robust performance across models and downstream tasks. Therefore, we choose $\gamma = 5$ and $r = 128$ as our default settings. We present our algorithm in Algorithm 1.

## 5 EXPERIMENTS

In this section, we conduct extensive experiments to evaluate the effectiveness of OwLore on multiple fine-tuning tasks. Details are provided below.

### 5.1 EXPERIMENTAL SETUP

**Pre-trained LLMs.** We choose multiple open-source LLMs that are widely used in research and practice, such as LLaMa2-7B (Touvron et al., 2023), LLaMa3-8B (Dubey et al., 2024), and Mistral-7B (Jiang et al., 2023).

**Fine-tuning Tasks.** We choose an extensive range of fine-tuning tasks aiming to provide a thorough evaluation of OwLore . Our fine-tuning tasks cover three categories: (i) **Commonsense Reasoning** (Hu et al., 2023), which includes 8 reasoning tasks including BoolQ (Clark et al., 2019), PIQA (Bisk et al., 2020), SIQA (Sap et al., 2019), HellaSWag (Zellers et al., 2019), WinoGrande (Sakaguchi et al., 2021), ARC-e (Clark et al., 2018), ARC-c (Clark et al., 2018), and OBQA (Mihaylov et al., 2018). (ii) **MT-Bench** (Zheng et al., 2024), a challenging multi-turn question set to assess the conversational and instruction-following abilities of models, including 8 common categories: writing, roleplay, extraction, reasoning, math, coding, STEM, and humanities. We apply GPT-3.5-turbo and GPT-4o as the judge for MT-Bench; (iii) **MMLU** (Hendrycks et al., 2020), a massive multitask test consisting of multiple-choice questions from various branches of knowledge. The test spans 57 tasks including elementary mathematics, US history, computer science, law, and more. We adopt the 5-shot setting for MMLU. For Commonsense Reasoning, all models are first fine-tuned on commonsense170k and then evaluated separately on different tasks, following Hu et al. (2023); For MT-Bench, we first fine-tune models on the Alpaca GPT-4 dataset (Peng et al., 2023) and then evaluate on MT-Bench following LISA. The results of MMLU are fine-tuned on the auxiliary training dataset and then evaluated on MMLU with 5 shots.

**PEFT Baselines.** We mainly consider four state-of-the-art baselines that are closely related to our approach: (i) *Full fine-tuning (Full FT)*: all parameters of pre-trained models are fine-tuned. Weights, gradients, and optimization states are maintained with full rank; (ii) *LoRA* Hu et al. (2021): LoRA introduces additional low-rank adaptors and only fine-tunes adaptors, while maintaining pre-trained weights frozen during training; (iii) *GaLore* Zhao et al. (2024): pre-trained LLMs are fine-tuned with

low-rank gradient projection. We follow Zhao et al. (2024) and set the rank level to 8 for both GaLore and LoRA in all fine-tuning tasks; (iv) *LISA* Pan et al. (2024): LISA is a sampling-based LLM fine-tuning method, which by default samples 2 layers to fine-tune with full rank at each iteration. Similar to our approach, both GaLore and LISA directly fine-tune pre-trained weights without adding additional adaptors.

To provide a comprehensive evaluation of our approach, we introduce two variants: (1) **OwLore**, the complete version of our method, and (2) **OwLore (Full-Rank)**, which only adopts OWS and excludes Gradient Low-Rank Projection. For a fair comparison, OwLore (Full-Rank) strictly adheres to the settings of LISA, unfreezing $\gamma = 2$ layers per iteration and updating them in full-rank. In contrast, OwLore leverages its memory efficiency by setting $\gamma = 5$ and $r = 128$.

**Hyperparameter Tuning.** Regarding the hyperparameters of the baselines, we have conducted extensive hyperparameter tuning for all baselines with LLaMa2-7B and LLaMa3-8B, and report the results with the best ones. For Mistral-7B, we directly use best hyperparameters of LLaMa3-8B. Specifically, for the learning rate, we performed a hyperparameter sweep over [1e-4, 3e-4, 7e-5, 5e-5, 1e-5, 5e-6] for each method. For GaLore, we tested several update frequencies for the subspace [50, 100, 200, 500] and found that 200 works best, consistent with GaLore's reports. To ensure a fair comparison, we followed GaLore's approach and set the rank level to 8 for GaLore and LoRA, resulting in approximately 24GB memory usage for all methods. Additionally, we thoroughly analyzed the effect of two hyperparameters, such as rank level and sampled layers, as shown in Figure 3, where our approach consistently demonstrates superior memory benefits. More configurations details are reported in Appendix C.

## 5.2 EXPERIMENTAL RESULTS

In this section, we present the empirical results of OwLore in comparison to other baseline methods.

**Commonsense Reasoning Benchmark.** We first evaluate with 8 commonsense reasoning tasks. The results are reported in Table 4. Overall, OwLore and OwLore (Full-Rank) consistently outperform Full FT and other PEFT baselines by a large margin across various LLMs, demonstrating the superiority of OwLore in LLM fine-tuning. We summarize our key observations below:

Table 4: Fine-tuning performance of LLaMa2-7B, Mistral-7B, and LLaMa3-8B with various approaches on commonsense reasoning datasets.

| Method | Mem. | BoolQ | PIQA | SIQA | HellaSwag | WinoGrande | ARC-e | ARC-c | OBQA | Avg. |
|---|---|---|---|---|---|---|---|---|---|---|
| | | | | | **LLaMa2-7B** | | | | | |
| Full FT | 61G | 87.3 | 79.5 | 32.7 | 56.7 | 80.2 | 78.5 | 49.0 | 40.8 | 63.1 |
| LoRA | 26G | 79.7 | 79.7 | 34.4 | 59.9 | 79.8 | 79.5 | 49.7 | 36.6 | 62.4 |
| GaLore | 36G | 81.8 | 79.4 | 32.9 | 60.7 | 79.6 | 79.8 | 49.4 | 37.6 | 62.7 |
| LISA | 24G | 82.0 | 79.9 | 33.5 | 59.7 | 79.6 | 80.4 | 51.1 | 38.8 | 63.1 |
| OwLore (Full-Rank) | 24G | 85.1 | 80.3 | 34.5 | 59.8 | 80.5 | 80.1 | 51.5 | 39.2 | 63.9 |
| OwLore | **23G** | 85.4 | 80.7 | 34.2 | 60.3 | 82.2 | 80.6 | 51.0 | 39.1 | **64.2** |
| | | | | | **LLaMa3-8B** | | | | | |
| Full FT | 61G | 86.8 | 82.5 | 33.6 | 63.1 | 83.1 | 83.6 | 53.3 | 37.4 | 65.4 |
| LoRA | 26G | 87.2 | 81.0 | 33.7 | 62.9 | 83.3 | 82.2 | 54.2 | 37.0 | 65.2 |
| GaLore | 36G | 85.0 | 81.8 | 33.1 | 61.9 | 83.6 | 83.5 | 52.8 | 38.8 | 65.1 |
| LISA | 24G | 87.3 | 81.6 | 33.7 | 61.7 | 83.6 | 82.7 | 54.4 | 38.8 | 65.5 |
| OwLore (Full-Rank) | 24G | 86.8 | 81.6 | 34.2 | 62.9 | 84.1 | 81.9 | 53.3 | 40.2 | 65.6 |
| OwLore | **23G** | 86.6 | 82.3 | 33.8 | 63.0 | 83.5 | 83.2 | 55.3 | 38.6 | **65.8** |
| | | | | | **Mistral-7B** | | | | | |
| Full FT | 61G | 86.5 | 84.3 | 33.5 | 65.1 | 87.1 | 83.8 | 57.5 | 41.2 | 67.4 |
| LoRA | 26G | 87.2 | 81.0 | 33.7 | 62.9 | 83.3 | 82.2 | 54.2 | 37.0 | 65.2 |
| GaLore | 36G | 84.8 | 82.5 | 34.4 | 63.5 | 85.6 | 82.5 | 53.9 | 37.8 | 65.6 |
| LISA | 24G | 84.7 | 82.9 | 33.4 | 64.2 | 85.8 | 83.4 | 54.4 | 40.5 | 66.2 |
| OwLore (Full-Rank) | 24G | 87.3 | 83.8 | 33.7 | 66.1 | 84.9 | 83.7 | 55.3 | 38.2 | 66.7 |
| OwLore | 23G | 87.8 | 83.9 | 34.0 | 66.4 | 85.6 | 84.1 | 57.9 | 40.4 | **67.5** |

① **OwLore approaches significantly outperform other efficient fine-tuning approaches by a large margin.** Applying OWS to LISA (i.e., OwLore (Full-Rank)) achieves a notable average accuracy boost over LISA on LLaMA2-7B, i.e., 0.8%. Moreover, the low-rank operation further improves the performance-memory trade-off of OwLore, achieving a 0.3% and 0.8% average accuracy gain with LLaMa2-7B and Mistral-7B, respectively.

② **OwLore approaches consistently outperform full fine-tuning across tasks on LLaMa.** We can observe that both OwLore and OwLore (Full-Rank) can outperform the performance of full fine-tuning with all models. LISA can match the performance of full fine-tuning for LLaMa models, whereas GaLore and LoRA perform no better than full fine-tuning. However, only OwLore is able to match the performance of full fine-tuning with Mistral-7B and all other baselines fail to do so.

③ **LLaMa3-8B consistently outperforms LLaMa2-7B on Commonsense Reasoning.** As the most advanced variant of LLaMa, LLaMa3-8B consistently outperforms its previous version. Interestingly, performance variance between different fine-tuning approaches of LLaMa3 is smaller than LLaMa2.

**MT-Bench.** We next evaluate OwLore on a more comprehensive benchmark, MT-Bench, featuring 80 high-quality, multi-turn questions designed to assess LLMs on 8 common categories. Results are presented in Table 5. We can observe that the benefits of OwLore over other PEFT approaches are more pronounced. All other baselines fail to match the performance of full fine-tuning on MT-Bench with scores below 6.0, whereas OwLore (Full-Rank) and OwLore both outperform the full fine-tuning by a large margin. OwLore (Full-Rank) significantly boosts the average score of LISA from 5.92 to 6.46 by solely applying OWS, highlighting the effectiveness of our outlier-inspired sampling.

Table 5: Fine-tuning performance of LLaMa2-7B with various approaches on MT-Bench using GPT-3.5-turbo as a judge.

| Method | Writing | Roleplay | Reasoning | Math | Coding | Extraction | STEM | Humanities | Avg. |
|---|---|---|---|---|---|---|---|---|---|
| Full-FT | 7.11 | 8.11 | 4.90 | 2.85 | 3.75 | 6.50 | 7.80 | 8.10 | 6.14 |
| LoRA | 7.21 | 7.05 | 4.95 | 3.25 | 3.90 | 5.70 | 7.90 | 7.65 | 5.95 |
| GaLore | 7.05 | 7.79 | 3.55 | 2.89 | 3.15 | 6.25 | 8.30 | 7.63 | 5.83 |
| LISA | 6.75 | 7.35 | 4.35 | 3.00 | 3.85 | 6.85 | 7.74 | 7.47 | 5.92 |
| OwLore (Full-Rank) | 7.53 | 8.00 | 4.93 | 3.25 | 4.53 | 6.33 | 8.50 | 8.57 | 6.46 |
| OwLore | 8.00 | 7.65 | 4.95 | 3.25 | 4.15 | 7.45 | 8.25 | 8.45 | **6.52** |

For MT-bench, we also evaluate the models using GPT-4 as the judge, which is a more commonly used choice. The results are shown in Table 6. As observed, the performance trend when using GPT-4 is very similar to that of GPT-3.5-turbo, although the scores evaluated by GPT-4 are generally lower. Notably, only OwLore (Full-Rank) and OwLore outperform full fine-tuning, with the complete version of OwLore achieving a significantly higher margin over full fine-tuning.

Table 6: Mean score of LLaMA-2-7B on MT-Bench fine-tuned by six fine-tuning methods over three seeds using GPT-4o as the judge.

| Model | Judge | Full-FT | LoRA | GaLore | LISA | OwLore (Full-Rank) | OwLore |
|---|---|---|---|---|---|---|---|
| LLaMa-2-7B | GPT-3.5-turbo | 6.14 | 5.95 | 5.83 | 5.92 | 6.46 | **6.52** |
| LLaMa-2-7B | GPT-4o | 4.91 | 4.58 | 4.73 | 4.81 | 4.95 | **5.10** |

Table 7: Fine-tuning performance of LLaMa2-7B with various approaches on MMLU benchmark.

| Method | Humanities | STEM | Social Sciences | Other | Avg. |
|---|---|---|---|---|---|
| Full-FT | 49.9 | 41.7 | 57.5 | 57.0 | 51.5 |
| LoRA | 46.1 | 40.8 | 56.6 | 56.2 | 49.9 |
| GaLore | 45.4 | 41.7 | 55.8 | 56.0 | 49.7 |
| LISA | 44.9 | 41.2 | 54.7 | 57.6 | 49.6 |
| OwLore (Full-Rank) | 49.1 | 41.3 | 58.8 | 59.1 | 52.1 |
| OwLore | 49.8 | 42.1 | 58.6 | 59.7 | **52.6** |

**MMLU Benchmark.** To draw a more solid conclusion, we also test another widely used benchmark, i.e., MMLU. The results are shown in Table 7. Our findings highlight that OwLore consistently outperforms Full FT, while other PEFT methods fall short of dense fine-tuning. Specifically, OwLore achieves an average score of 52.6, demonstrating significant improvements across various domains such as Humanities, STEM, Social Sciences, and Others. These results underscore OwLore's efficacy beyond full fine-tuning while maintaining superior memory efficiency.

## 5.3 FINE-TUNING MEMORY USAGE

Thanks to its layerwise sampling and low-rank characteristics, OwLore significantly improves the memory efficiency of LLM fine-tuning. To verify this, we report the memory cost of various approaches when used to fine-tune LLaMa2-7B, with a token batch size of 1, as shown in Figure 3.

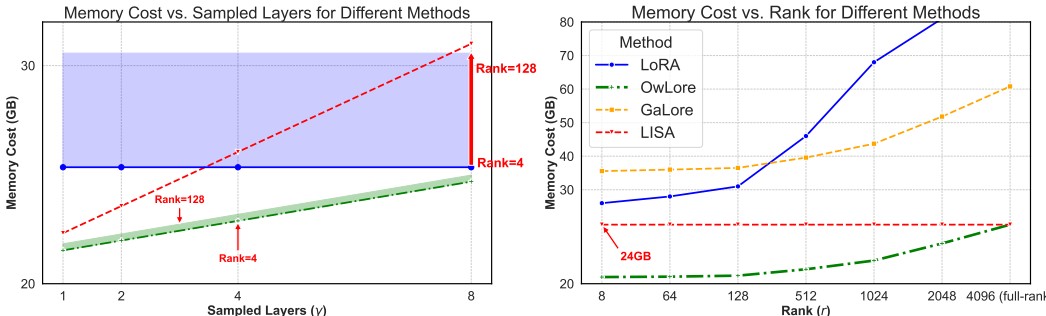

Figure 3: Fine-tuning memory usage of using various with LLaMa2-7B. **Left:** varying sampled layers. In this scenario, we also vary the rank of LoRA and OwLore from 4 to 128 to provide a comprehensive analysis. OwLore consistently demonstrates superior memory efficiency across all configurations. Notably, LISA's memory advantage over LoRA diminishes as the number of sampled layers increases. **Right:** varying ranks. The sampled layer of LISA and OwLore is set as $\gamma = 2$.

On the one hand, the low-rank nature of OwLore allows us to unfreeze more layers without a substantial increase in memory cost compared to LISA. As illustrated in Figure 3-Left, when increasing $\gamma$ from 1 to 8, LISA exhibits a notable memory growth from 23GB to 32GB, whereas OwLore's memory cost slightly increases from 21GB to 25GB. Compared to LoRA with $r = 4$, OwLore facilitates training with a much higher rank ($r = 128$) while still maintaining a lower memory cost. On the other hand, Figure 3-Right demonstrates that OwLore enables high-rank training without significantly compromising memory efficiency, in stark contrast to LoRA. It is important to note that we do not utilize the layer-wise weight update technique used in GaLore for the memory measurement, hence the memory cost of GaLore is higher than reported in GaLore.

We further break down the memory usage during LLM fine-tuning, presenting the results in Figure 4-Left. For this analysis, $\gamma$ is set to 2 for both LISA and OwLore, and $r$ is set to 8 for both LoRA and OwLore. LoRA incurs a substantial activation memory cost, although its optimizer and gradient memory requirements are relatively small. In contrast, LISA's optimizer memory cost is large because each layer is trained in full rank, yet it benefits from a small activation memory cost. OwLore effectively combines the advantages of both methods, inheriting the small activation memory of LISA while significantly reducing the optimizer memory requirement. Notably, this benefit allows OwLore to fine-tune LLaMa2-7B with only 22GB of memory, demonstrating its superior memory efficiency.

## 5.4 TRAINING LOSS CURVE

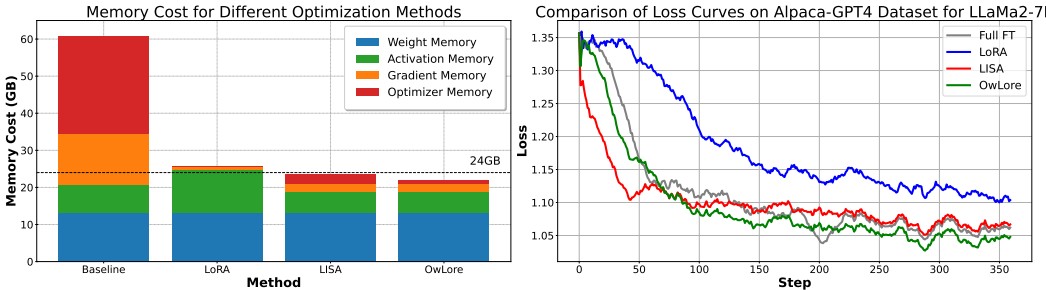

Figure 4: **Left:** Mmeory breakdown of various methods using LLaMa2-7B. **Right:** Fine-tuning loss of LLaMA2-7B on Alpaca GPT-4 dataset using various methods.

The training loss curve is an effective way to understand the training dynamics of various methods. Following LISA, we present fine-tuning loss curves of LLaMa2-7B on the Alpaca-GPT4 dataset using Full FT, LoRA, LISA, and OwLore in Figure 4-Right. At first glance, methods that directly fine-tune pre-trained weights (i.e., LISA and OwLore) can better mimic the training landscape of full fine-tuning, compared to LoRA.

It is worth noting that while OwLore initially falls short of LISA in the early phase of training, it gradually catches up after 60 iterations and eventually outperforms LISA with a lower loss. We conjecture that the underlying reason here is that the low-rank update of OwLore is less accurate than the full-rank update of LISA at the beginning. However, as training progresses, OwLore keeps updating the subspace, leading to an optimal one.

## 6 CONCLUSION

In this paper, we study the sampling-based LLM fine-tuning, where at each iteration, only a few layers are sampled and fine-tuned, instead of the whole model. Specifically, we delve into recently-proposed LISA (Pan et al., 2024) and unveil two shortcomings that constrain its memory-performance trade-off: (1) The middle layers of LISA are sampled uniformly, which can result in suboptimal performance. (2) The sampled layers of LISA are fine-tuned in a full-rank manner, causing a significant memory increase as the number of sampled layers increases. To solve these problems, we introduce **OwLore**, a novel fine-tuning method that assigns higher sampling probabilities to these outlier-rich layers. This innovative technique enhances fine-tuning performance while maintaining higher memory efficiency compared to traditional full-rank fine-tuning. The memory efficiency of OwLore could be further improved by incorporating Low-Rank gradient projection. Combining sampling-based fine-tuning with gradient low-rank projection not only enhances the performance-memory trade-off of sampling-based fine-tuning but also boosts the effectiveness of gradient low-rank projection in LLM fine-tuning, Our experiments across various architectures, including LLaMa2, LLaMa3, and Mistral, demonstrate that OwLore achieves significant performance improvements while maintaining higher memory efficiency compared to traditional full-rank fine-tuning.

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

## A PSEUDOCODE OF GALORE

Following we present the pseudocode for Galore (Zhao et al., 2024). As part of the Owlore algorithm, the low-rank updating nature of Galore could help to further improve the memory efficiency.

---

**Algorithm 2:** GaLore

---

**Input:** A layer weight matrix $W \in \mathbb{R}^{m \times n}$ with $m \leq n$. Step size $\eta$, scale factor $\alpha$, decay rates $\beta_1, \beta_2$,
         rank $r$, subspace change frequency $T$.
**Output:** Updated weight matrix $W_t$.
Initialize first-order moment $M_0 \in \mathbb{R}^{n \times r} \leftarrow 0$
Initialize second-order moment $V_0 \in \mathbb{R}^{n \times r} \leftarrow 0$
Initialize step $t \leftarrow 0$
**while** *convergence criteria not met* **do**
    $G_t \in \mathbb{R}^{m \times n} \leftarrow -\nabla_W \phi_t(W_t)$
    **if** $t \bmod T = 0$ **then**
        $U, S, V \leftarrow \text{SVD}(G_t)$
        $P_t \leftarrow U[:, :r]$                             ▷ *Initialize left projector as $m \leq n$*
    **else**
        $P_t \leftarrow P_{t-1}$                               ▷ *Reuse the previous projector*
    $R_t \leftarrow P_t^\top G_t$                          ▷ *Project gradient into compact space*

    **Update** $(R_t)$ **by Adam**
    $M_t \leftarrow \beta_1 \cdot M_{t-1} + (1 - \beta_1) \cdot R_t$
    $V_t \leftarrow \beta_2 \cdot V_{t-1} + (1 - \beta_2) \cdot R_t^2$
    $M_t \leftarrow M_t / (1 - \beta_1^t)$
    $V_t \leftarrow V_t / (1 - \beta_2^t)$
    $N_t \leftarrow M_t / (\sqrt{V_t} + \epsilon)$

    $\tilde{G}_t \leftarrow \alpha \cdot P N_t$                      ▷ *Project back to original space*
    $W_t \leftarrow W_{t-1} + \eta \cdot \tilde{G}_t$
    $t \leftarrow t + 1$
**return** $W_t$

---

## B HYPERPARAMETER ANALYSIS

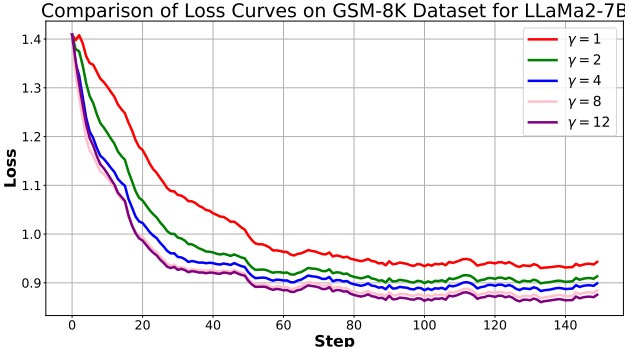

Figure 5: Fine-tuning loss of LLaMA2-7B using method OwLore on the GSM-8K dataset with various sampled layers.

$\tau$ is the key hyperparameter to obtain the outlier ratio and sampling layers $\gamma$ is also crucial to OwLore To obtain intuitive and empirical guidance on these hyperparameter choices, we conduct ablation studies using LLaMA2-7B models with the GSM-8K dataset and report the results below.

Table 8: GSM scores for different $\tau$ values

| Setting | $\tau = 3$ | $\tau = 5$ | $\tau = 7$ | $\tau = 9$ | $\tau = 11$ | $\tau = 13$ | $\tau = 15$ | $\tau = 17$ | $\tau = 19$ |
|---|---|---|---|---|---|---|---|---|---|
| GSM Scores | 19.18 | 19.41 | 20.04 | 20.62 | 21.15 | 20.24 | 20.17 | 20.47 | 19.79 |

We found that mid-range values of $\tau$, such as 9, 11 and 13, generally lead to better performance. This may stem from the fact that the outliers screened by these values are more indicative of heavy-tailed properties. By default, we choose $\tau = 13$ for all experiments of OwLore.

As for the sampling layer $\gamma$, it is not surprising that performance improves consistently with the sampling of more layers. OwLore outperforms LISA with less memory usage across all sampling layer counts. This is attributed to OwLore's allocation of higher sampling probabilities to layers abundant in outliers, combined with its efficient low-rank gradient updating technique.

The training curve across different values of $\gamma$ is depicted in Figure 5. Notably, fine-tuning with a higher $\gamma$ leads to faster convergence and lower loss.

## C  TRAINING CONFIGURATIONS OF OWLORE

Table 9: Hyperparamters used of OwLore for fine-tuning LLaMa2-7B, LLaMa3-8B, and Mistral-7B on the Commonsense Reasoning Benchmark.

| Hyperparameter | LLaMa2-7B | LLaMa3-8B | Mistral-7B |
|---|---|---|---|
| Batch Size | 16 | 16 | 16 |
| Max. Sequence Length | 512 | 512 | 512 |
| Learning Rate | 3e-4 | 7e-5 | 3e-5 |
| Schedular | linear | linear | linear |
| Training Epoch | 1 | 1 | 1 |
| Warmup Steps | 0 | 0 | 0 |
| dtype | bfloat16 | bfloat16 | bfloat16 |

Table 10: Hyperparamters used of OwLore for fine-tuning LLaMa2-7B on various benchmarks.

| Benchmarks | Commonsense Reasoning | MT-Bench | MMLU | GSM8K |
|---|---|---|---|---|
| Train Samples | 170K | 52K | 99.8K | 7.4K |
| Test Samples | 22.4K | Alpaca-GPT4 (3.3K) | 14K | 1.3K |
| Batch Size | 16 | 16 | 16 | 16 |
| Max_length | 512 | 512 | 512 | 512 |
| Training Epoch | 1 | 1 | 1 | 1 |
| Learning Rate | 3e-4 | 3e-4 | 3e-4 | 3e-4 |

