# OpenReview forum: "OwLore: Outlier-weighed Layerwise Sampled Low-Rank Projection for Memory-Efficient LLM Fine-tuning"
_ICLR.cc/2025/Conference — Submitted to ICLR 2025_

### Official Review · Reviewer_2nYd · 2024-10-29

**Soundness:** 3
**Presentation:** 3
**Contribution:** 3
**Rating:** 5
**Confidence:** 4

**Summary:**

This paper proposes OwLore, a novel method for updating LLM layers using a sampling-based strategy. Specifically, layers with a higher concentration of outliers have an increased probability of being updated. OwLore also incorporates gradient low-rank projection to further reduce memory costs. Extensive experiments across various architectures on commonsense reasoning, MMLU, and MT-Bench demonstrate the effectiveness of OwLore.

**Strengths:**

1. The method is well-motivated, with a novel and intuitive outlier-based sampling strategy.

2. Experiments using several LLMs, including Llama2, Llama3 and Mistral, across various tasks like commonsense reasoning, MMLU and MT-Bench confirm the effectiveness of the approach, boosting performance over baselines without increased memory cost.

3. The code for OwLore is available, supporting reproducibility and further exploration.

**Weaknesses:**

1. OwLore may lead to increased time costs, as the outlier ratio for layers must be computed with each update. However, the experiments do not include a comparison of time costs, which seems unfair to baseline methods, especially PEFT methods that do not use sampling. Even with sampling based methods like LISA, its random sampling strategy will likely lead to less time cost than OwLore. Including time cost metrics would provide a more balanced comparison and highlight the efficiency trade-offs of OwLore.

2. In Figure 4.4, the finetuning loss curve is not converging, with an even sharper drop in the last few optimization steps, making the analysis in this section less convincing. Furthermore, a similar pattern is also observed in Figure 5.

**Questions:**

1. In Figure 2, it appears that the last (bottom) layer does not have a high outlier score in OwLore, while the LISA paper indicates that the bottom layer should have a higher importance score and is consistently optimized. What might account for this discrepancy between LISA and OwLore?

2. In Line 215, the phrase "rich-get-richer" likely means that layers sampled and fine-tuned more frequently will, in turn, accumulate more outliers. This creates a feedback loop where layers with more outliers are sampled more often, which then leads to even more outliers in those layers. Could the authors clarify if this effect is intended?

---

> ### Author Response · Authors · 2024-11-22
>
> ### **Response to Reviewer 2nYd**
>
> We would like to thank the reviewer for their thoughtful comments and for recognizing the motivation, reproducibility, and universality of our work across several baselines. We address the concerns below.
>
> **W1: OwLore may lead to increased time cost.**
>
> - We appreciate your concern. We would like to clarify that the outlier scores are computed only once before the fine-tuning process begins, not with each update. This preprocessing step is performed prior to fine-tuning and does not introduce overhead during the training iterations.
> For example, in our experiments with the LLaMA2-7B model, the computation of outlier scores takes approximately 73 secs. In contrast, the total fine-tuning time is about 1.6 hours, making the outlier score computation a negligible portion (approximately 1.2%) of the overall training time.
>
>
> - For each training step, whether it is LISA or OwLore, the layers are sampled based on a given probability distribution. Therefore, the time consumption is exactly the same (approximately 0.06s).
> We will update the manuscript to include a detailed analysis of the time costs, providing a fair comparison with baseline methods and highlighting the efficiency trade-offs of OwLore.
>
> **W2: In Figure 4.4, the finetuning loss curve is not converging, with an even sharper drop in the last few optimization steps, making the analysis in this section less convincing. Furthermore, a similar pattern is also observed in Figure 5.**
>
> - Thank you for bringing this to our attention. We have provided complete loss curves that demonstrate full convergence and updated the manuscript in Figure 4-right and Figure 5.
>
> - Besides, the complete loss values are presented in the table below. The results show that OwLore not only converges faster but also achieves a lower final loss compared to the baseline methods.
>
> | Method | 0 | 29 | 59 | 89 | 119 | 149 | 179 | 209 | 239 | 269 | 299 | 329 | 359 |
> |--------|------|------|------|------|------|------|------|------|------|------|------|------|------|
> | LoRA   | 1.3563 | 1.3413 | 1.3054 | 1.2501 | 1.1889 | 1.1581 | 1.1571 | 1.1314 | 1.1366 | 1.1278 | 1.1265 | 1.1163 | 1.1037 |
> | FT     | 1.3563 | 1.2879 | 1.1307 | 1.1171 | 1.1179 | 1.0837 | 1.0896 | 1.0531 | 1.0756 | 1.0764 | 1.0688 | 1.0504 | 1.0619 |
> | LISA   | 1.3563 | 1.1545 | 1.1248 | 1.1043 | 1.1001 | 1.0880 | 1.0854 | 1.0851 | 1.0802 | 1.0821 | 1.0727 | 1.0581 | 1.0669 |
> | OwLore | 1.3563 | 1.2329 | 1.1478 | 1.0997 | 1.0836 | 1.0664 | 1.0642 | 1.0622 | 1.0608 | 1.0621 | 1.0529 | 1.0341 | 1.0479 |
>
>
>
> **Q1: In Figure 2, it appears that the last (bottom) layer does not have a high outlier score in OwLore, while the LISA paper indicates that the bottom layer should have a higher importance score and is consistently optimized. What might account for this discrepancy between LISA and OwLore?**
> - Thank you for this insightful observation. We would like to clarify that the discrepancy arises from differing definitions of the "bottom layer" between the two methods.
> In the LISA paper, the terms ’top’ and ’bottom’ layers refer to the embedding layer and the LLM head layer, respectively, rather than the first and last Transformer blocks.
> For the transformer block layers, LISA applies uniform random sampling during fine-tuning.
>
>  - In OwLore, we also acknowledge the significance of the embedding layer. Similar to LISA, we fine-tune the embedding layer without sampling because of its fundamental impact on the model's performance. For transformer layers, OwLore assigns different sampling probabilities based on each layer's outlier score, which reflects its importance. Layers with higher outlier ratios are sampled more frequently, allowing us to focus fine-tuning efforts where they have the most effect. Therefore, there is no discrepancy between LISA and OwLore in the bottom layer, which is the LLM head layer.
>
>
> **Q2:Clarify the phrase "rich-get-richer"**
>
> - We appreciate the opportunity to clarify this point. In our context, the "rich-get-richer" phenomenon refers to layers with higher initial outlier scores being sampled more frequently for fine-tuning, which leads to these layers being better trained. However, this does not imply that these layers will accumulate more outliers over time as a result of the fine-tuning process.
> Our intention is to prioritize layers that are inherently more significant—those with higher initial outlier ratios—for fine-tuning. By allocating more training resources to these layers,  the feedback loop enhances their learning and, consequently, the overall model performance.
>
>     We have revised the wording in the manuscript to make this concept clearer and to avoid any misunderstanding.

---

> > ### Comment · Reviewer_2nYd · 2024-11-25
> >
> > Thank you for your response; it addresses most of my concerns.
> >
> > However, I noticed a potential discrepancy that needs clarification. You mentioned that "the outlier scores are computed only once before the fine-tuning process begins." Yet, Algorithm 1 includes a sampling period K, which suggests that outlier scores are computed every K iterations, rather than only once. This inconsistency is quite confusing and should be addressed. Therefore, I will retain my current score.

---

> > > ### Author Response · Authors · 2024-11-27
> > > **Response from Authors**
> > >
> > > Dear Reviewer 2nYd,
> > >
> > > Thank you for your thoughtful feedback and for acknowledging that we have addressed most of your concerns. We appreciate the opportunity to clarify the potential discrepancy you highlighted regarding the computation of outlier scores.
> > >
> > > To clarify, the outlier scores are indeed computed only once before the fine-tuning process begins, as mentioned in our response. The sampling period K in Algorithm 1 refers to the periodicity of sampling layers for fine-tuning based on the pre-computed outlier scores. The outlier scores themselves are not recomputed every K iterations; instead, the sampling probability is pre-computed before fine-tuning. We apologize for any confusion caused by the phrasing in Algorithm 1, and we have updated the text in our revision for clarity.
> > >
> > > Given that this clarification resolves the final point of confusion, we kindly ask you to reconsider your score, as we have addressed all your concerns in full. We believe this work presents a significant and well-substantiated contribution, and your updated evaluation would mean a great deal to us.
> > >
> > > Thank you again for your valuable feedback and consideration.
> > >
> > > Best regards,
> > >
> > > The Authors

---

> ### Author Response · Authors · 2024-11-25
>
> Dear Reviewer 2nYd,
>
> We sincerely appreciate your insightful review, which has been helped in enhancing the quality of our work. As we approach the conclusion of the discussion phase, we would be happy to answer if you have any more concerns.
>
> Warmest regards,
> Authors

---

> ### Comment · Area_Chair_9Lcs · 2024-11-25
> **Please engage with author responses**
>
> The rebuttal period is coming to an end.

---

### Official Review · Reviewer_gXqP · 2024-11-03

**Soundness:** 3
**Presentation:** 3
**Contribution:** 2
**Rating:** 6
**Confidence:** 4

**Summary:**

The paper "OWLORE: Outlier-Weighed Layerwise Sampled Low-Rank Projection for LLM Fine-Tuning" presents a memory-efficient fine-tuning approach for large language models (LLMs). The proposed method, OwLore, introduces an outlier-weighted sampling strategy, focusing on layers with a higher concentration of outliers, which are considered more critical for fine-tuning (though more insights can be provided on why? See weakness). Unlike previous methods such as LISA, OwLore selectively fine-tunes layers based on their outlier distribution, and to further enhance memory efficiency, it uses a gradient low-rank projection for these layers. Experimental results show that OwLore outperforms both full fine-tuning and baseline methods in terms of memory efficiency and accuracy across benchmarks, including Commonsense Reasoning and MT-Bench.

**Strengths:**

The authors present a straightforward method to reduce the memory costs of fine-tuning large language models. They propose a more effective layer sampling strategy than the uniform approach used in the LISA baseline, selecting pretrained layers based on a targeted sampling method. Additionally, to further increase the number of pretrained layers involved in fine-tuning without raising memory costs, they incorporate GaLore-based gradient low-rank projection. While the method itself may not be highly innovative, the combination of these techniques is intriguing. Experimental results indicate consistent improvements across various datasets and baseline methods.

**Weaknesses:**

1. Importance of Outlier Weights for Fine-Tuning: Why are outlier weights more important for fine-tuning? Lines 91-94 lack supporting evidence. The statement that “we assign higher sampling probabilities to layers with a greater concentration of outliers, essentially forming a rich-get-richer phenomenon, substantially improving the fine-tuning performance” requires additional justification.

2. Unclear Rationale for the Choice of Outlier Score: The rationale behind the choice of outlier score is unclear.

3. The results presented are not fully convincing without detailed hyperparameter settings for the baseline methods, including the number of iterations for each method. It is particularly unclear why full-model fine-tuning is less effective than the proposed approach, which uses gradient low-rank projection and fine-tunes only five layers instead of the full model. Claims such as “our method outperforms full fine-tuning by a large margin” are potentially misleading, as the gains reported are relatively modest and may fall within standard deviation. Further clarification is needed on why OwLore (Full-Rank) is less effective than OwLore with gradient low-rank projection. Additionally, how does OwLore (Full-Rank) with a gamma setting applied to five layers compare directly to the proposed method? Memory costs should not increase significantly and warrant examination.

4. Comparative Performance of LoRA and Iteration Counts: How does LoRA with rank 16 perform? It would also be useful to know the number of iterations used for LoRA compared to other methods, as it might perform better with longer training durations.

5. It would be more informative to compare with GaLore, with the rank set to 128, similar to OwLore with gradient low-rank projection.

6. LISA Performance and Suggested Ablation Study: Since OwLore with gradient low-rank projection uses five layers, it would be insightful to examine how LISA performs with five layers under the same conditions. If LISA is expected to require more memory, consider conducting an ablation study on OwLore using gradient low-rank projection but without the outlier score, employing uniform sampling across five layers.

7. I request the authors to run the experiments on table 4 for 5 different seeds and provide the standard deviation. Furthermore, please provide the statistical significance test on the results.

**Questions:**

See above.

**Details Of Ethics Concerns:**

Not needed.

---

> ### Author Response · Authors · 2024-11-22
>
> ### **Response to Reviewer gXqP [1/4]**
>
> Thank you for your time and effort in reviewing our work. We are pleased that you found our method intriguing and that it demonstrates consistent improvements across various settings.
>
> **W1,W2: Why are outlier weights more important for fine-tuning? The rationale for the choice of outlier score?**
>
>
> - The importance of outliers, defined as activations [4,5] or weights [1,6] whose magnitudes are significantly larger than the others in LLMs, has been widely studied and verified. For instance, [4] first discovered the existence of outlier activations, and showed that setting these outlier feature dimensions to zero decreases top-1 attention softmax probability mass by more than 20% and degrades validation perplexity by 600-1000% despite them only making up about 0.1% of all input features. After that, numerous algorithms have been proposed to compress LLMs while taking care of those activation outliers [4,5,6,8,9] and weight outliers [1,6,10].
>
>     Given the pivotal role of outliers in LLMs, we argue that it is essential to consider outlier ratios when selecting layers for fine-tuning. Our intuition here is that layers with a higher proportion of outliers should be prioritized for fine-tuning, as these outlier weights tend to receive larger gradients, leading to greater magnitudes. This indicates that they contribute more significantly to the loss function and, consequently, are more critical for optimizing the model's performance.
>
> - To further validate this approach, we introduce a new baseline: OWS (reverse). This variant assigns lower sampling probabilities to layers with a higher proportion of outliers. As expected, OWS (reverse) performs the worst among the tested fine-tuning strategies, reinforcing our intuition about the importance of outlier-weighted prioritization in achieving better results.
>
>
>     | Method          | MMLU | BoolQ | PIQA | SIQA | HellaSwag | WinoGrande | ARC-e | ARC-c | OBQA | avg. |
>     |-----------------------------------------|---------------|-------------|------|-------|------|------|-----------|------------|-------|-------|
>     | OwLore-Reverse          | 49.4 | 81.9  | 77.8 | 33.4 | 59.1      | 80.1       | 79.3  | 50.2  | 38.2 | 61.0 |
>     | Galore           | 49.6 | 81.8  | 79.4 | 32.9 | 60.7      | 79.6       | 79.8  | 49.4  | 37.6 | 61.2 |
>     | LISA     | 49.6 | 82.0  | 79.9 | 33.5 | 59.7      | 79.6       | 80.4  | 51.1  | 38.8 | 61.6 |
>     | OwLore         | 52.6 | 85.4  | 80.7 | 34.2 | 60.3      | 82.2       | 80.6  | 51.0  | 39.1 | 62.9 |
>
>
>
>
>     [1] Yin, Lu, You Wu, Zhenyu Zhang, Cheng-Yu Hsieh, Yaqing Wang, Yiling Jia, Gen Li et al. "Outlier weighed layerwise sparsity (owl): A missing secret sauce for pruning llms to high sparsity." ICML 2024.
>
>     [2] Gromov, A., Tirumala, K., Shapourian, H., Glorioso, P. and Roberts, D.A., 2024. The unreasonable ineffectiveness of the deeper layers. arXiv preprint arXiv:2403.17887.
>
>     [3] Men, X., Xu, M., Zhang, Q., Wang, B., Lin, H., Lu, Y., Han, X. and Chen, W., 2024. Shortgpt: Layers in large language models are more redundant than you expect. arXiv preprint arXiv:2403.03853.
>
>     [4] Dettmers, Tim, Mike Lewis, Younes Belkada, and Luke Zettlemoyer. "Gpt3. int8 (): 8-bit matrix multiplication for transformers at scale." Advances in Neural Information Processing Systems 35 (2022): 30318-30332.
>
>     [5] Xiao, G., Lin, J., Seznec, M., Wu, H., Demouth, J. and Han, S., 2023, July. Smoothquant: Accurate and efficient post-training quantization for large language models. In International Conference on Machine Learning (pp. 38087-38099). PMLR.
>
>     [6] Kim, S., Hooper, C., Gholami, A., Dong, Z., Li, X., Shen, S., Mahoney, M.W. and Keutzer, K., 2023. Squeezellm: Dense-and-sparse quantization. arXiv preprint arXiv:2306.07629.
>
>     [7] Lin, J., Tang, J., Tang, H., Yang, S., Chen, W.M., Wang, W.C., Xiao, G., Dang, X., Gan, C. and Han, S., 2024. AWQ: Activation-aware Weight Quantization for On-Device LLM Compression and Acceleration. Proceedings of Machine Learning and Systems, 6, pp.87-100.
>
>     [8] Lee, C., Jin, J., Kim, T., Kim, H. and Park, E., 2024, March. Owq: Outlier-aware weight quantization for efficient fine-tuning and inference of large language models. In Proceedings of the AAAI Conference on Artificial Intelligence (Vol. 38, No. 12, pp. 13355-13364).
>
>     [9] Wei, X., Zhang, Y., Zhang, X., Gong, R., Zhang, S., Zhang, Q., Yu, F. and Liu, X., 2022. Outlier suppression: Pushing the limit of low-bit transformer language models. Advances in Neural Information Processing Systems, 35, pp.17402-17414.
>
>     [10] Sun, M., Liu, Z., Bair, A. and Kolter, J.Z., 2023. A simple and effective pruning approach for large language models. arXiv preprint arXiv:2306.11695.

---

> ### Author Response · Authors · 2024-11-22
>
> ### **Response to Reviewer gXqP [2/4]**
>
> **W1: The statement that “we assign higher sampling probabilities to layers with a greater concentration of outliers, essentially forming a rich-get-richer phenomenon, substantially improving the fine-tuning performance” requires additional justification.**
>
>
> - In our context, the "rich-get-richer" phenomenon refers to prioritizing layers that inherently have higher initial outlier ratios. By assigning higher sampling probabilities to these layers, we ensure they are sampled more frequently for fine-tuning. This approach allocates more training resources to the most significant layers, enhancing their learning and, consequently, improving the overall model performance.
>
> - We would like to clarify that this phenomenon does not imply that these layers will accumulate more outliers over time due to the fine-tuning process due to the smalle changes of weights during fine-tuning. Rather, we leverage the existing outlier distribution to guide our layer selection, focusing on layers that are more influential in the model's performance from the outset.
>
>
>
>
> **W3 (1): The results presented are not fully convincing without detailed hyperparameter settings for the baseline methods, including the number of iterations for each method.**
>
> - For all baselines and OwLore, we trained models for the same number of iterations, with a batch size of 16, ensuring consistency in the number of training iterations. We used the following shared parameters for all methods discussed in our paper.
>
>     | Hyperparameter        | LLaMa2-7B | LLaMa3-8B | Mistral-7B |
>     |-----------------------|-----------|-----------|------------|
>     | Batch Size           | 16        | 16        | 16         |
>     | Max. Sequence Length | 512       | 512       | 512        |
>     | Scheduler            | linear    | linear    | linear     |
>     | Training Epoch       | 1         | 1         | 1          |
>     | Warmup Steps         | 0         | 0         | 0          |
>     | dtype                | bfloat16  | bfloat16  | bfloat16   |
>
> - We have shared the hyperparameters of different fine-tuning approaches in Section 4.1 lines 310 to 320. As for the learning rate, we performed a hyperparameter sweep over [1e-4, 3e-4, 7e-5, 5e-5, 1e-5, 5e-6] for each method. For GaLore, we tested several update frequencies for the subspace [50, 100, 200, 500] and found that 200 works best, consistent with GaLore's reports. To ensure a fair comparison, we followed GaLore's approach and set the rank level to 8 for GaLore and LoRA, resulting in approximately 24GB memory usage for all methods. Additionally, we thoroughly analyzed the effect of two hyperparameters, such as rank level and sampled layers, as shown in Figure 3, where our approach consistently demonstrates superior memory benefits.
>
> **W3 (2): It is particularly unclear why full-model fine-tuning is less effective than the proposed approach, which uses gradient low-rank projection and fine-tunes only five layers instead of the full model.**
>
>
> - Thank you for highlighting this important observation. We would like to clarify that full fine-tuning is not always the most effective baseline. It often suffers from the "learns more and forgets more" phenomenon, where the model may overfit to new data and forget previously acquired knowledge [1]. This issue can lead to diminished performance and generalization capabilities.
>
> - Due to this reason, many PEFT methods have been shown to outperform full fine-tuning such as LISA [2], PISSA [3], DoRA [4] etc. For instance，LISA employs importance sampling, achieving superior performance compared to full-parameter fine-tuning in certain scenarios.
>
> - Our approach further enhances the effectiveness of LISA by  focusing on layers with a higher concentration of outliers and efficiently managing gradients through low-rank projection. Therefore, it is not surprising to see it can consistently outperform full fine-tuning across various benchmarks.
>
>
>     [1] Lora learns less and forgets less, TMLR 2024.
>
>     [2] Pan, R., Liu, X., Diao, S., Pi, R., Zhang, J., Han, C. and       Zhang, T., 2024. LISA: Layerwise Importance Sampling for Memory-     Efficient Large Language Model Fine-Tuning. NeurIPS 2024.
>
>     [3] Meng, Fanxu, Zhaohui Wang, and Muhan Zhang. "Pissa:
>     Principal singular values and singular vectors adaptation of
>     large language models." arXiv preprint arXiv:2404.02948 (2024).
>
>     [4] Liu, S.Y., Wang, C.Y., Yin, H., Molchanov, P., Wang, Y.C.F.,     Cheng, K.T. and Chen, M.H., 2024. Dora: Weight-decomposed low- rank adaptation. arXiv preprint arXiv:2402.09353.

---

> ### Author Response · Authors · 2024-11-22
>
> ### **Response to Reviewer gXqP [3/4]**
>
> **W3 (3): Claims such as “our method outperforms full fine-tuning by a large margin” are potentially misleading, as the gains reported are relatively modest and may fall within standard deviation.**
>
> - We appreciate the opportunity to clarify our results and address your concerns. We respectfully disagree with the assertion that the gains are relatively modest and may fall within the standard deviation. Our experimental results demonstrate consistent and meaningful improvements over full fine-tuning across multiple benchmarks.
>
>     Specifically:
>
>     - Commonsense Reasoning with LLaMA2-7B: OwLore achieves an accuracy of 64.2%, representing an increase of 1.1% points over full fine-tuning.
>
>     - MT-Bench: OwLore scores 6.52, which is an increase of 0.38 points or a 6.16% relative improvement over full fine-tuning.
>
> - In the context of large language models and challenging benchmarks, even improvements of 1% can be significant. These gains are particularly noteworthy given that our method also reduces memory consumption and computational requirements compared to full fine-tuning.
>
>
>
>
> **W3 (4): Further clarification is needed on why OwLore (Full-Rank) is less effective than OwLore with gradient low-rank projection.**
>
> - OwLore: This is the full version of our approach, utilizing gradient low-rank projection (GaLore). This technique allows fine-tuning of more layers at each step without increasing memory costs. Specifically, we fine-tune 5 layers at each step, with each layer updated in a low-rank space using 128 ranks.
>
> - OwLore (Full-Rank): This is the full-rank variant of OwLore. With the same memory allocation, this approach can fine-tune only 2 layers, making it less effective compared to OwLore's low-rank implementation.
>
>
> **W3 (5): Additionally, how does OwLore (Full-Rank) with a gamma setting applied to five layers compare directly to the proposed method? Memory costs should not increase significantly and warrant examination**
>
> - Thank you for your question. Regarding OwLore (Full-Rank) with a gamma setting applied to five layers, we acknowledge that the memory cost increases from **23.49G** to **27.04G**, representing approximately a 15% increase. However, this increase is not negligible, particularly in memory-constrained environments where efficient deployment is critical.
>
>     | Method | Memory |
>     | -------- |-------- |
>     | OwLore       |  23.49G    |
>     | OwLore (Full-Rank)     |   27.04G   |
>
>
>
> **W4: Comparative Performance of LoRA and Iteration Counts: How does LoRA with rank 16 perform? It would also be useful to know the number of iterations used for LoRA compared to other methods, as it might perform better with longer training durations.**
>
>
> - We clarify that all approaches in our paper are trained with the same number of iterations. To address your concern, we conducted additional experiments comparing LoRA of 16 ranks with OwLore across different numbers of training epochs on LLaMa2-7B GSM8K. The results are summarized in the table below
>
>
>     | Method  |  LoRA (Rank=16) | OwLore |
>     | -------- | -------- | -------- |
>     | Epoch = 1     |    18.2   | 23.9 |
>     | Epoch = 2     |    19.8   | 24.3 |
>     | Epoch = 3     |    20.5   | 25.8 |
>
> While additional training epochs lead to improvements for LoRA, it still falls short of OwLore consistently with all training epochs.  Specifically, after three epochs, OwLore achieves a score of 25.8, which is a 5.3 percentage point higher than LoRA with Rank 16.
>
>
>
> **W5: It would be more informative to compare with GaLore, with the rank set to 128, similar to OwLore with gradient low-rank projection.**
>
>
>
> - We report the results LLaMa2-7B on GSM8K in the table below，where r is the rank number and γ is the sampled layers. Notably, with the same rank of 128, OwLore can still outperform GaLore by a good margin while significantly reduce memory usage from 36.5G to 22G, even OwLore only samples 2 layers at each time step.
>
>
>     | **Method**   | **Setting**           | **Result (GSM8K score/memory)** |
>     |--------------|-----------------------|--------------------------------|
>     | *Galore*     | `r=128, γ=32`         | 18.2 / 36.5G                  |                 |
>     | *OwLore*     | `r=128, γ=2`         | **21.9 / 22G**                |

---

> ### Author Response · Authors · 2024-11-22
>
> ### **Response to Reviewer gXqP [4/4]**
>
> **W6: LISA Performance and Suggested Ablation Study: Since OwLore with gradient low-rank projection uses five layers, it would be insightful to examine how LISA performs with five layers under the same conditions. If LISA is expected to require more memory, consider conducting an ablation study on OwLore using gradient low-rank projection but without the outlier score, employing uniform sampling across five layers.**
>
> - Thank you for your suggestion. We conducted experiments to compare LISA and OwLore under the specified conditions, i.e., fine-tuning with five layers with gradient low-rank projection. The results are reported in the following table. We can clearly see that OwLore outperforms LISA with a 1% average improvement. This ablation study confirms that our outlier-weighted sampling (OWS) is crucial for superior performance. Simply applying gradient low-rank projection with uniform layer sampling is less effective than our approach.
>
>
>     | Model                                   | Sample Layers | Galore Used | MMLU | BoolQ | PIQA | SIQA | HellaSwag | WinoGrande | ARC-e | ARC-c | OBQA | avg. |
>     |-----------------------------------------|---------------|-------------|------|-------|------|------|-----------|------------|-------|-------|------|------|
>     | LISA   | 5             | Yes         | 49.6 | 81.9  | 80.1 | 33.3 | 60.1      | 81.4       | 80.7  | 51.2  | 39.2 | 61.9 |
>     | OwLore     | 5             | Yes         | **52.6** | **85.4**  | **80.7** | **34.2** | **60.3**      | **82.2**       | 80.6  | 51.0  | 39.1 | **62.9** |
>
>
>
>
>
> **W7: I request the authors to run the experiments on Table 4 for 5 different seeds and provide the standard deviation. Furthermore, please provide the statistical significance test on the results.**
>
>
>
> - As requested, we conducted experiments using 5 different seeds and reported the corresponding standard deviations. However, due to time constraints, we were unable to complete all the planned experiments. Instead, we prioritized experiments with LISA and OwLore to demonstrate the effectiveness of our proposed approach.
>
>     For MT-Bench, we provided the results evaluated using GPT-4o.
>
>
>     | Model      | Method   | BoolQ       | PIQA        | SIQA        | HellaSwag   | WinoGrande   | ARC-e       | ARC-c       | OBQA        |
>     |:-----------|:---------|:------------|:------------|:------------|:------------|:-------------|:------------|:------------|:------------|
>     | LLaMa2-7B  | LISA     | 81.9 ± 0.22 | 79.6 ± 0.26 | 33.6 ± 0.11 | 59.6 ± 0.09 | 79.5 ± 0.16  | 80.3 ± 0.13 | 51.1 ± 0.11 | 39.2 ± 0.12 |
>     | LLaMa2-7B  | OwLore   | **85.3 ± 0.19** | **80.8 ± 0.29** | **34.2 ± 0.14** | **60.2 ± 0.11** | **82.4 ± 0.18**  | **80.8 ± 0.14** | **51.1 ± 0.12** | **39.6 ± 0.15** |
>     | LLaMa3-8B  | LISA     | 87.2 ± 0.18 | 81.8 ± 0.19 | 33.6 ± 0.09 | 61.7 ± 0.10 | 83.5 ± 0.12  | 82.6 ± 0.12 | 54.1 ± 0.15 | 39.2 ± 0.10 |
>     | LLaMa3-8B  | OwLore   | 86.7 ± 0.19 | **82.3 ± 0.14** | **33.6 ± 0.10** | **62.9 ± 0.13** | **83.5 ± 0.11**  | **83.4 ± 0.10** | **55.5 ± 0.13** | **39.4 ± 0.11** |
>     | Mistral-7B | LISA     | 84.9 ± 0.21 | 82.7 ± 0.21 | 33.4 ± 0.11 | 64.4 ± 0.14 | 85.7 ± 0.16  | 83.6 ± 0.11 | 54.3 ± 0.10 | 40.5 ± 0.14 |
>     | Mistral-7B | OwLore   | **88.0 ± 0.24** | **84.0 ± 0.23** | **33.9 ± 0.11** | **66.4 ± 0.16** | **85.8 ± 0.09**  | **84.1 ± 0.15** | **57.8 ± 0.14** | **40.5 ± 0.13** |
>
>
>     | Method   | MT-Bench          |
>     |---------|----------------|
>     | LISA    | 4.92 ± 0.14    |
>     | OwLore  | **5.14 ± 0.16**    |
>
> - Additionally, we perform an independent samples t-test to assess the statistical significance of the performance difference between OwLore and LISA. For example, in LLaMa2-7B model, the t-test yields a t-statistic of -11.36 and a p-value of 3.41e-06, indicating that the performance improvements of OwLore over LISA are statistically significant.
>
>
>
>     | Model  | t-statistic | p-value |
>     | -------- | -------- | -------- |
>     | LLaMa2-7B     | -11.36     | 3.41e-06     |
>     | LLaMa3-8B     | -3.87     | 0.0047     |
>     | Mistral-7B     | -13.46     | 9.32e-07     |

---

> ### Author Response · Authors · 2024-11-25
>
> Dear Reviewer gXqP,
>
> We are truly grateful for your thoughtful comments, which has significantly contributed to the improvement of our work. As we approach the end of the discussion phase, please don't hesitate to let us know if you have any further concerns, and we would be more than happy to address them.
>
> Kind regards,
>
> The Authors

---

> ### Comment · Area_Chair_9Lcs · 2024-11-25
> **Please engage with author responses**
>
> The rebuttal period is coming to an end. Have the new experiments satisfied your issues with the paper?

---

> > ### Comment · Reviewer_gXqP · 2024-11-26
> > **Thanks for the response**
> >
> > I appreciate the response. I will increase my score accordingly.

---

> > > ### Author Response · Authors · 2024-11-27
> > > **We sincerely appreciate your response and the increase in your score.**
> > >
> > > Dear Reviewer gXqP,
> > >
> > > We sincerely thank you for appreciating our response. Your positive feedback and support mean a great deal to us!
> > >
> > > Best regards,
> > >
> > > The Authors

---

### Official Review · Reviewer_WvN5 · 2024-11-03

**Soundness:** 3
**Presentation:** 3
**Contribution:** 3
**Rating:** 5
**Confidence:** 4

**Summary:**

The paper introduces Outlier-weighed Layerwise Sampled Low-Rank Projection (OwLore), a memory-efficient fine-tuning method for large language models. OwLore improves performance by focusing on layers with higher outlier distributions and selectively fine-tuning those layers. It also employs gradient low-rank projection to enhance efficiency further. Experimental results show that OwLore outperforms baseline methods, achieving significant accuracy gains on benchmarks like Commonsense Reasoning, MMLU, and MT-Bench.

**Strengths:**

- The proposed method achieves better performance with lower memory requirements.
 - The proposed OwLore includes a novel evaluation of outlier of weights in each layer, hence enable an effective selection algorithm.
 - Incorporating GaLore, further decrease the memory requirements.

**Weaknesses:**

- The only contribution of the paper is the metric of how to evaluate the outlier in weights, which is kind of marginal.
- The paper should elaborate why this kind of evaluation is useful, where the idea comes from.
- The author should not only just focus the experiments on how much memory is saved, but also, how much the hyperparameter in OwLore affects the performance.
- The paper should provide some insights from the methods that why combine selection and GaLore con boost performance. Why from the results the two methods are compatible.


A follow-up comment:

In my opinion, a research paper being accepted should propose a novel, interesting methodology, give a clear explanation of where this idea comes from and why it results in such a form, and also demonstrate the effectiveness of the designs and how they relate with intuition.
Maybe the author could detail why the outlier matters so much and why they use such a function to evaluate this.

For the hyperparameter ablations, maybe the tau? and also gamma? Does the method excel at different combinations of hyperparameters? This is one key experiment and the result will show whether the method is superior.

An ablation study can be added to the paper to decompose the contribution of the designs, or just the authors share the insight for this.

Overall, given all this, the paper is not that kind of paper qualified to be accepted, and there are a lot of experiments and analyses that need to be included in the paper, and also the authors' explanation of the results. In my opinion, you can only do so many things with methodology design, and the thing that really matters are those ideas from the experiments from whatever designs which this paper lacks.

I am open to rebuttal and any further demonstrations to change my score.

**Questions:**

N/A

---

> ### Author Response · Authors · 2024-11-22
>
> ### **Response to Reviewer WvN5 [1/3]**
> **W1: The only contribution of the paper is the metric of how to evaluate the outlier in weights, which is kind of marginal.**
> - First and foremost, we emphasize that our contribution extends beyond introducing a metric for evaluating outlier distributions. The primary contribution of our paper lies in **identifying the limitations of layerwise-sampling-based LLM fine-tuning (LISA) and proposing OwLore to address these limitations and significantly enhance performance**. Initially, LISA [1] employs importance sampling by selectively fine-tuning only a few layers at each step while keeping the remaining layers frozen. This layerwise sampling strategy allows LISA to achieve substantial memory savings during fine-tuning while outperforming both LoRA and even full-parameter fine-tuning in certain scenarios.
>
> - However, we observe two potential limitations of LISA: **(1)** The middle layers of LISA are sampled uniformly, which can result in suboptimal performance, as LLM’s layers are not equally important [2,3,4]. Table 1 in our submission also confirms this; **(2)** The sampled layers of LISA are fine-tuned in a full-rank manner, causing a significant memory increase as the number of sampled layers increases.
> - To address these two limitations, we propose OwLore, which leverages **Outlier-Weighed Sampling (OWS)**, and **GaLore** to address the above two limitations, respectively. OWL strategically assigns higher sampling probabilities to layers with more outliers, such that layers with more outlier weights can be fine-tuned more frequently. GaLore, on the other hand, reduces the memory cost of fine-tuning by projecting a full gradient to a low-rank subspace, which allows us to activate more layers without linearly increasing memory costs. It is important to highlight that demonstrating the efficacy of low-rank gradient on layerwise sampling for LLM fine-tuning is also new, not only because no previous work has explored this, but also because this combination achieves a synergistic effect where the whole is greater than the sum of its parts. We believe all the above aspects are meaningful contributions to the community.
>
>
> **W2: The paper should elaborate why this kind of evaluation is useful, and where the idea comes from.**
> - The importance of outliers, defined as activations [5,6] or weights [2,7] whose magnitudes are significantly larger than the others in LLMs, has been widely studied and verified. For instance, [5] first discovered the existence of outlier activations, and showed that setting these outlier feature dimensions to zero decreases top-1 attention softmax probability mass by more than 20% and degrades validation perplexity by 600-1000% despite them only making up about 0.1% of all input features. After that, numerous algorithms have been proposed to compress LLMs while taking care of those activation outliers [5,6,8,9,10] and weight outliers [2,7,11].
>
> - Given the pivotal role of outliers in LLMs, we argue that it is essential to consider outlier ratios when selecting layers for fine-tuning. Our intuition here is that layers with a higher proportion of outliers should be prioritized for fine-tuning, as these outlier weights tend to receive larger gradients, leading to greater magnitudes. This indicates that they contribute more significantly to the loss function and, consequently, are more critical for optimizing the model's performance.
>
> - To further validate this approach, we introduce a new baseline: OWS (reverse). This variant assigns lower sampling probabilities to layers with a higher proportion of outliers. As expected, OWS (reverse) performs the worst among the tested fine-tuning strategies, reinforcing our intuition about the importance of outlier-weighted prioritization in achieving better results.
>
>
>     | Method          | MMLU | BoolQ | PIQA | SIQA | HellaSwag | WinoGrande | ARC-e | ARC-c | OBQA | avg. |
>     |-----------------------------------------|---------------|-------------|------|-------|------|------|-----------|------------|-------|-------|
>     | OwLore-Reverse          | 49.4 | 81.9  | 77.8 | 33.4 | 59.1      | 80.1       | 79.3  | 50.2  | 38.2 | 61.0 |
>     | Galore           | 49.6 | 81.8  | 79.4 | 32.9 | 60.7      | 79.6       | 79.8  | 49.4  | 37.6 | 61.2 |
>     | LISA     | 49.6 | 82.0  | 79.9 | 33.5 | 59.7      | 79.6       | 80.4  | 51.1  | 38.8 | 61.6 |
>     | OwLore         | 52.6 | 85.4  | 80.7 | 34.2 | 60.3      | 82.2       | 80.6  | 51.0  | 39.1 | 62.9 |

---

> ### Author Response · Authors · 2024-11-22
>
> ### **Response to Reviewer WvN5 [2/3]**
> **W3, Follow-up Comment: The author should not only just focus the experiments on how much memory is saved, but also, on how much the hyperparameter in OwLore affects the performance. For the hyperparameter ablations, does the method excel at different combinations of hyperparameters? This is one key experiment and the result will show whether the method is superior.**
>
> - We answer these two questions together here. We confrim that OwLore excels at different combinations of hyperparameters. The results are summarized in the below table for W4. As shown, OwLore consistently demonstrates superior performance over LISA's and GaLore's all combinations of hyperparameters.
> The only exception is the setting of "r=128, γ=1", where we only fine-tune one layer at each step. However, even in this extremely memory-efficient configuration, we can outperform LISA (r=full, γ=8) with a notable **11G memory saving**.
>
> **W4: The paper should provide some insights from the methods that why combine selection and GaLore can boost performance. Why from the results the two methods are compatible?**
>
> - Thanks for your great quesiton! Combining selection (LISA) and GaLore can address the limitation of each algorithm, achieving a synergistic effect where the whole is greater than the sum of its parts. Let us elaborate in detail.
>
> - One limitation of LISA is that, while it achieves improved performance, its memory cost increases linearly with the number of fine-tuned layers. This is because LISA's each layer is fine-tuned in full rank. On the other hand, GaLore's limitation lies in its mediocre fine-tuning performance, which does not improve significantly as the rank increases. We illustrate these limitations in the following table using LLaMA2-7B on GSM8K. Here, r is the rank level, γ is number layers selected for fine-tuning, the results are report with the "Accuracy/Memory" format.
>
> - Notably, combining GaLore with LISA significantly reduces the memory cost compared to LISA alone-reducing from 36G to 27G with "r=full, γ=12"-while achieving a significant 6.1% accuracy gain. The success of this combination lies in the fact that GaLore allows LISA to update the sampled layers in a memory-efficient low-rank space. This enables fine-tuning of more layers without a dramatic increase in memory consumption.
>
>
>
>
>     |      Method          |             |        |  Setting      |        |      |
>     |------------------|-----------------|------------------|------------------|------------------|------------------|
>     | *LISA*       | r=full, γ=1     | r=full, γ=2      | r=full, γ=4      | r=full, γ=8      | r=full, γ=12     |
>     |                  | 16.8/23G        | 18.8/25G         | 19.8/27G         | 19.9/32G         | 21.7/36G         |
>     |  *GaLore*     |  r=8, γ=32     | r=16, γ=32       | r=32, γ=32       | r=64, γ=32       | r=128, γ=32      |
>     |                  | 19.1/35.6G      | 18.8/35.6G       | 18.4/35.8G       | 18.7/36.0G       | 18.2/36.5G       |
>     | *OwLore*     | r=128, γ=1      | r=128, γ=2       | r=128, γ=4       | r=128, γ=8       | r=128, γ=12      |
>     |                  | 20.0/21G        | 21.9/22G         | 23.5/23G         | 25.7/25G         | **27.8/27G**         |

---

> ### Author Response · Authors · 2024-11-22
>
> ### **Response to Reviewer WvN5 [3/3]**
>
> **Follow-up Comment: An ablation study can be added to the paper to decompose the contribution of the designs, or just the authors share the insight for this.**
>
> - The respective contributions are already evaluated separately in our paper. In Table 4-6, the OwLore (full-rank) is ''LISA+OWS'' where we sample 2 layers at each step (same as LISA); and OwLore essentially represents LISA+OWS+GaLore where we sample 5 layers, and each layer is trained with 128 ranks. We report the results here as well for your convenience.
>
>     **Table: Fine-tuning performance of LLaMa2-7B**
>     | Method                | Mem. | BoolQ | PIQA | SIQA | HellaSwag | WinoGrande | ARC-e | ARC-c | OBQA | Avg. |
>     |-----------------------|------|-------|------|------|-----------|------------|-------|-------|------|------|
>     | Full FT              | 61G  | 87.3  | 79.5 | 32.7 | 56.7      | 80.2       | 78.5  | 49.0  | 40.8 | 63.1 |
>     | LoRA                 | 26G  | 79.7  | 79.7 | 34.4 | 59.9      | 79.8       | 79.5  | 49.7  | 36.6 | 62.4 |
>     | GaLore               | 36G  | 81.8  | 79.4 | 32.9 | 60.7      | 79.6       | 79.8  | 49.4  | 37.6 | 62.7 |
>     | LISA                 | 24G  | 82.0  | 79.9 | 33.5 | 59.7      | 79.6       | 80.4  | 51.1  | 38.8 | 63.1 |
>     | OwLore (Full-Rank)   | 24G  | 85.1  | 80.3 | 34.5 | 59.8      | 80.5       | 80.1  | 51.5  | 39.2 | 63.9 |
>     | OwLore               | 23G  | 85.4  | 80.7 | 34.2 | 60.3      | 82.2       | 80.6  | 51.0  | 39.1 | 64.2 |
>
>     We hope our response addresses all your concerns. Please feel free to let us know if there are any additional points you would like us to clarify.
>
>
>     **Reference**
>
>     [1] Pan, R., Liu, X., Diao, S., Pi, R., Zhang, J., Han, C. and Zhang, T., 2024. LISA: Layerwise Importance Sampling for Memory-Efficient Large Language Model Fine-Tuning. NeurIPS 2024.
>
>     [2] Yin, Lu, You Wu, Zhenyu Zhang, Cheng-Yu Hsieh, Yaqing Wang, Yiling Jia, Gen Li et al. "Outlier weighed layerwise sparsity (owl): A missing secret sauce for pruning llms to high sparsity." ICML 2024.
>
>     [3] Gromov, A., Tirumala, K., Shapourian, H., Glorioso, P. and Roberts, D.A., 2024. The unreasonable ineffectiveness of the deeper layers. arXiv preprint arXiv:2403.17887.
>
>     [4] Men, X., Xu, M., Zhang, Q., Wang, B., Lin, H., Lu, Y., Han, X. and Chen, W., 2024. Shortgpt: Layers in large language models are more redundant than you expect. arXiv preprint arXiv:2403.03853.
>
>     [5] Dettmers, Tim, Mike Lewis, Younes Belkada, and Luke Zettlemoyer. "Gpt3. int8 (): 8-bit matrix multiplication for transformers at scale." Advances in Neural Information Processing Systems 35 (2022): 30318-30332.
>
>     [6] Xiao, G., Lin, J., Seznec, M., Wu, H., Demouth, J. and Han, S., 2023, July. Smoothquant: Accurate and efficient post-training quantization for large language models. In International Conference on Machine Learning (pp. 38087-38099). PMLR.
>
>     [7] Kim, S., Hooper, C., Gholami, A., Dong, Z., Li, X., Shen, S., Mahoney, M.W. and Keutzer, K., 2023. Squeezellm: Dense-and-sparse quantization. arXiv preprint arXiv:2306.07629.
>
>     [8] Lin, J., Tang, J., Tang, H., Yang, S., Chen, W.M., Wang, W.C., Xiao, G., Dang, X., Gan, C. and Han, S., 2024. AWQ: Activation-aware Weight Quantization for On-Device LLM Compression and Acceleration. Proceedings of Machine Learning and Systems, 6, pp.87-100.
>
>     [9] Lee, C., Jin, J., Kim, T., Kim, H. and Park, E., 2024, March. Owq: Outlier-aware weight quantization for efficient fine-tuning and inference of large language models. In Proceedings of the AAAI Conference on Artificial Intelligence (Vol. 38, No. 12, pp. 13355-13364).
>
>     [10] Wei, X., Zhang, Y., Zhang, X., Gong, R., Zhang, S., Zhang, Q., Yu, F. and Liu, X., 2022. Outlier suppression: Pushing the limit of low-bit transformer language models. Advances in Neural Information Processing Systems, 35, pp.17402-17414.
>
>     [11] Sun, M., Liu, Z., Bair, A. and Kolter, J.Z., 2023. A simple and effective pruning approach for large language models. arXiv preprint arXiv:2306.11695.

---

> ### Author Response · Authors · 2024-11-25
>
> Dear Reviewer WvN5,
>
> We sincerely appreciate your thoughtful feedback, which has greatly contributed to improving our work. As we approach the end of the discussion phase, please feel free to share any additional concerns, we would be more than happy to address them.
>
> Kind regards,
>
> The Authors

---

> ### Comment · Area_Chair_9Lcs · 2024-11-25
> **Please engage with author responses**
>
> The rebuttals are coming to an end. You expressed a willingness to change your score, and I'm sure the authors are hoping for you to consider their response.

---

> ### Comment · Reviewer_WvN5 · 2024-11-25
> **Response**
>
> Given the rebuttal, I don't really feel convinced.
>
> for the novelty, although the author clarified where the idea is from, but it is still Galore+OWS, and the only interesting part is the metric. This is not enough for publishing.
>
> for the hyperparameter ablation, I am curious about how the hyperparameters affect the task performance, not the memory.
>
> for the overall improvement, the avg improvement is dominated by BoolQ, this is a biased evaluation.
>
> I would maintain my current score.

---

> ### Author Response · Authors · 2024-11-27
> **Clarification of  your concerns**
>
> Thank you for your continued feedback and for taking the time to evaluate our rebuttal. We would like to respectfully emphasize that all of your concerns were thoroughly addressed in our previous response, and we are confident in the robustness and significance of our contributions.
>
> - **First**, we respectfully disagree with your assertion that OWS (our proposed metric) is insufficient for publication. The primary objective of our work is to advance sampling-based LLM fine-tuning, specifically through LISA, where the most crucial aspect is the methodology for layer sampling. **OWS introduces a novel approach by selecting layers with more outliers rather than relying on uniform sampling, which is both reasonable and demonstrably effective.** Importantly, OWS (denoted as "OwLore (Full-Rank)") alone provides significant improvements in LISA's fine-tuning performance. While GaLore is included as a secondary enhancement, OWS itself represents a standalone contribution that addresses key limitations in sampling strategies for LLM fine-tuning.
>
> - **Second**, as shown in our updated results, OwLore consistently outperforms baseline methods across diverse hyperparameter configurations. The following table directly addresses concerns about the robustness of our approach under varying conditions.
>
>    |      Method          |             |        |  Setting      |        |      |
>    |------------------|-----------------|------------------|------------------|------------------|------------------|
>    | *LISA*       | r=full, γ=1     | r=full, γ=2      | r=full, γ=4      | r=full, γ=8      | r=full, γ=12     |
>    |                  | 16.8/23G        | 18.8/25G         | 19.8/27G         | 19.9/32G         | 21.7/36G         |
>    |  *GaLore*     |  r=8, γ=32     | r=16, γ=32       | r=32, γ=32       | r=64, γ=32       | r=128, γ=32      |
>    |                  | 19.1/35.6G      | 18.8/35.6G       | 18.4/35.8G       | 18.7/36.0G       | 18.2/36.5G       |
>    | *OwLore*     | r=128, γ=1      | r=128, γ=2       | r=128, γ=4       | r=128, γ=8       | r=128, γ=12      |
>    |                  | **20.0/21G**        | **21.9/22G**        | **23.5/23G**         | **25.7/25G**         | **27.8/27G**         |
>
> - **Third**, we would like to reiterate that OwLore achieves consistent and substantial improvements across a range of datasets. While BoolQ demonstrates a larger gain due to the task's alignment with our method's strengths, this does not diminish the broad and consistent performance improvements observed across other datasets. These results in the following table confirm the generalizability and effectiveness of our approach, far beyond any single dataset. The datasets where OwLore outperforms LISA are marked in bold below.
>
>    **Table: Fine-tuning performance of LLaMa2-7B**
>    | Method                | Mem. | BoolQ | PIQA | SIQA | HellaSwag | WinoGrande | ARC-e | ARC-c | OBQA | Avg. |
>    |-----------------------|------|-------|------|------|-----------|------------|-------|-------|------|------|
>    | LISA                 | 24G  | 82.0  | 79.9 | 33.5 | 59.7      | 79.6       | 80.4  | 51.1  | 38.8 | 63.1 |
>    | OwLore (Full-Rank)   | 24G  | **85.1**  | **80.3** | **34.5** | **59.8**      | **80.5**       | 80.1  | **51.5**  | **39.2** | **63.9** |
>    | OwLore               | 23G  | **85.4**  | **80.7** | **34.2** | **60.3**     | **82.2**       | **80.6**  | 51.0  | **39.1** | **64.2** |
>
>
> We firmly believe that our responses and additional evidence fully address the concerns raised and substantiate the merit of our work. We hope this clarifies any remaining misunderstandings.

---

### Official Review · Reviewer_wV28 · 2024-11-03

**Soundness:** 3
**Presentation:** 2
**Contribution:** 3
**Rating:** 6
**Confidence:** 4

**Summary:**

This paper proposes the combination of the following two techniques:
- Outlier-Weighed Sampling (OWS): a heuristics for stochastic & selective layer wise fine-tuning.
- GaLore: a low-rank-update optimization method family proposed by Zhao _et al._ (2024).

Comparing against Full FT, GaLore, and LoRA, both OWS and OwLore (OWS + Galore) are competitive while keeping peak memory usage lower than 1/2 of Full FT.

**Strengths:**

The paper presents an effective and memory-efficient fine-tuning method that is competitive. Empirical evaluation spans multiple datasets.

**Weaknesses:**

1. Presentation seems a bit confusing. This paper first introduces OWS as a technique but in Tables 4-7 it is listed as OwLore (full-rank). Selective layer freezing / GaLore are distinct techniques. And I think the presentation would be clearer if their respective contributions are separately evaluated.
2. Paper could be strengthened if more LISA-D experiments are included. From Table 1 it seems the even simpler heuristics of favoring shallower layers is already effective - is Eq 2 really necessary? Including LISA-D as one of the baselines in Tables 4-7 will help answer this question.

**Questions:**

What are the # of trainable parameters of methods compared in this paper?

---

> ### Author Response · Authors · 2024-11-22
>
> ### **Response to Reviewer wV28 [1/2]**
> We would first like to thank you for your time and effort in reviewing our work. We are glad that you have found our efficient fine-tuning method to be competitive. We would like to address the weaknesses pointed out by you one by one as follows:
>
> **W1: Presentation seems a bit confusing. This paper first introduces OWS as a technique but in Tables 4-7 it is listed as OwLore (full-rank). Selective layer freezing / GaLore are distinct techniques. And I think the presentation would be clearer if their respective contributions are separately evaluated.**
> - We thank you for your suggestions, which helped improve our paper's presentation.
> First of all, please allow us to reiterate the contribution of our paper. The overarching goal of our paper is to advance the current layerwise-sampling-based LLM fine-tuning, which was introduced in LISA [1]. Specifically, LISA adopts importance sampling for LLM fine-tuning, where only a couple of layers are sampled to be fine-tuned at each step, and keep the rest of the layers frozen. Such layerwise sampling allows LISA to save largely fine-tuning memory while outperforming LoRA and even full-parameter fine-tuning under certain settings.
>
> - However, we observe two potential limitations of LISA (1) The middle layers of LISA are sampled uniformly, which can result in suboptimal performance, as LLM’s layers are not equally important [2,3,4]. Table 1 in our submission also confirms this; (2) The sampled layers of LISA are fine-tuned in a full-rank manner, causing a significant memory increase as the number of sampled layers increases.
>
> - To address these two limitations, we propose OwLore, which leverages Outlier-Weighed Sampling (OWS), and GaLore to address the above two limitations, respectively. OWL strategically assigns higher sampling probabilities to layers with more outliers, such that layers with more outlier weights can be fine-tuned more frequently. GaLore, on the other hand, reduces the memory cost of fine-tuning by projecting a full gradient to a low-rank subspace, which allows us to activate more layers without linearly increasing memory costs.
>
> - **It is important to highlight** that it is meaningful to combine LISA with GaLore, as this combination achieves a synergistic effect where the whole is greater than the sum of its parts. Specifically, as we demonstrated below with LLaMa2-7B on GSM8K.  Here, r is the rank level, γ is number layers selected for fine-tuning, the results are report with the "Accuracy/Memory" format. Notably, combining GaLore with LISA significantly reduces the memory cost compared to LISA alone-reducing from 36G to 27G with "r=full, γ=12"-while achieving a significant 6.1% accuracy gain. The success of this combination lies in the fact that GaLore allows LISA to update the sampled layers in a memory-efficient low-rank space. This enables fine-tuning of more layers without a dramatic increase in memory consumption.
>
>
>     |      Method          |             |        |  Setting      |        |      |
>     |------------------|-----------------|------------------|------------------|------------------|------------------|
>     |  *Galore*     |  r=8, γ=32     | r=16, γ=32       | r=32, γ=32       | r=64, γ=32       | r=128, γ=32      |
>     |                  | 19.1/35.6G      | 18.8/35.6G       | 18.4/35.8G       | 18.7/36.0G       | 18.2/36.5G       |
>     | *LISA*       | r=full, γ=1     | r=full, γ=2      | r=full, γ=4      | r=full, γ=8      | r=full, γ=12     |
>     |                  | 16.8/23G        | 18.8/25G         | 19.8/27G         | 19.9/32G         | 21.7/36G         |
>     | *OwLore*     | r=128, γ=1      | r=128, γ=2       | r=128, γ=4       | r=128, γ=8       | r=128, γ=12      |
>     |                  | 20.0/21G        | 21.9/22G         | 23.5/23G         | 25.7/25G         | **27.8/27G**         |
>
>
>
>
> - In addition, we fully agree with your great suggestions. We have modified our presentation following your suggestions. Concretely, we changed Section 3 into “Limitations of Layerwise Importance Sampled AdamW (LISA)”, where we introduce LISA’s algorithm and its two limitations. In Section 4, we propose OwLore which leverages OWS and GaLore to enhance LISA’s performance and memory efficiency.

---

> ### Author Response · Authors · 2024-11-22
>
> ### **Response to Reviewer wV28 [2/2]**
>
> **W2: Paper could be strengthened if more LISA-D experiments are included. From Table 1 it seems the even simpler heuristics of favoring shallower layers is already effective - is Eq 2 really necessary? Including LISA-D as one of the baselines in Tables 4-7 will help answer this question.**
>
> - We thank you for your question. Yes, Eq 2 is a more effective approach than LISA-D. We have included LISA-D with LLaMa2-7B. We can see that while it outperforms LISA in most cases, it falls short of OwLore (Full-Rank) and OwLore consistently. Please note that the only difference between LISA-D and OwLore (Full-Rank) is the sampling approach.
>
>     | Method                | Mem. | BoolQ | PIQA | SIQA | HellaSwag | WinoGrande | ARC-e | ARC-c | OBQA | Avg.  |
>     |-----------------------|------|-------|------|------|-----------|------------|-------|-------|------|-------|
>     | Full FT               | 61G  | 87.3  | 79.5 | 32.7 | 56.7      | 80.2       | 78.5  | 49.0  | 40.8 | 63.1  |
>     | LoRA                  | 26G  | 79.7  | 79.7 | 34.4 | 59.9      | 79.8       | 79.5  | 49.7  | 36.6 | 62.4  |
>     | GaLore                | 36G  | 81.8  | 79.4 | 32.9 | 60.7      | 79.6       | 79.8  | 49.4  | 37.6 | 62.7  |
>     | LISA                  | 24G  | 82.0  | 79.9 | 33.5 | 59.7      | 79.6       | 80.4  | 51.1  | 38.8 | 63.1  |
>     | LISA-D               | 24G  | 85.1  | 79.9 | 33.8  | 59.8      | 79.7       |  80.0 | 51.3  | 38.4 | 63.5  |
>     | OwLore (Full-Rank)    | 24G  | 85.1  | 80.3 | 34.5 | 59.8      | 80.5       | 80.1  | 51.5  | 39.2 | 63.9  |
>     | OwLore                | 23G  | 85.4  | 80.7 | 34.2 | 60.3      | 82.2       | 80.6  | 51.0  | 39.1 | **64.2** |
>
> **Q1: What are the # of trainable parameters of methods compared in this paper?**
> - Thank you for bringing up this question. The **overall number of trainable parameters** is the same for **full-parameter fine-tuning**, **LISA**, **GaLore**, and **OwLore**, as all parameters of their base model are trainable. The only exception is **LoRA**, where the number of trainable parameters is smaller than full-parameter fine-tuning due to it only trains the low-rank adaptors.
>
> - However, the **memory usage during fine-tuning** is not determined solely by the total number of parameters. Other factors also play a significant role, such as:
>   - **Trainable parameters at each step**: LISA and OwLore update only a few layers at each step.
>   - **Optimizer states**: GaLore and OwLore update these in a low-rank subspace.
>
> Below, we present a table showing the trainable parameters for each training step calculated using different methods. Please note that the **trainable parameters** do not fully reflect the memory usage of approaches that use **gradient low-rank projection**, such as **GaLore** and **OwLore**. Even if their entire parameters are updated, their **optimizer states** are updated in a low-rank subspace. As the memory cost of optimizer states is typically twice as large as the parameters, their memory usage is significantly smaller.
>
>    **Table: Trainable Parameters Per Step in LLaMa2-7B**
>    | Method | Full FT | LoRA | GaLore| LISA | OwLore | OwLore (Full-Rank)|
>    | -------- | -------- | -------- | -------- |-------- |-------- |-------- |
>    | Trainable Parameters     |   6.7B   |   4.2M   |  6.7B  |  333.4M   |  602.9M |  333.4M   |
>
>    **Reference**
>
>    [1] Pan, R., Liu, X., Diao, S., Pi, R., Zhang, J., Han, C. and Zhang, T., 2024. LISA: Layerwise Importance Sampling for Memory-Efficient Large Language Model Fine-Tuning. NeurIPS 2024.
>
>    [2] Yin, Lu, You Wu, Zhenyu Zhang, Cheng-Yu Hsieh, Yaqing Wang, Yiling Jia, Gen Li et al. "Outlier weighed layerwise sparsity (owl): A missing secret sauce for pruning llms to high sparsity." ICML 2024.
>
>    [3] Gromov, A., Tirumala, K., Shapourian, H., Glorioso, P. and Roberts, D.A., 2024. The unreasonable ineffectiveness of the deeper layers. arXiv preprint arXiv:2403.17887.
>
>    [4] Men, X., Xu, M., Zhang, Q., Wang, B., Lin, H., Lu, Y., Han, X. and Chen, W., 2024. Shortgpt: Layers in large language models are more redundant than you expect. arXiv preprint arXiv:2403.03853.

---

> ### Author Response · Authors · 2024-11-25
>
> Dear Reviewer wV28,
>
> We are truly thankful for your insightful feedback, which has significantly enhanced our work. As we approach the conclusion of the discussion phase, please feel free to share any additional concerns, and we would be more than happy to address them.
>
> Kind regards,
>
> The Authors

---

> ### Comment · Area_Chair_9Lcs · 2024-11-25
> **Please engage with author responses**
>
> Rebuttals are coming to an end. I'm aware that the authors submitted theirs late in the discussion period, but I hope you at least confirm you've read them.

---

> ### Comment · Reviewer_wV28 · 2024-11-26
> **Thanks for the response!**
>
> I appreciate the additional experiments. Please incorporate them in the revised manuscript.

---

### Meta-Review · Area_Chair_9Lcs · 2024-12-18

**Metareview:**

This paper proposes a combination of existing efficiency methods with a novel proposed method of selecting layers to tune based on their outlier weights.

Pros:
Novel metric for outlier weights.
Several experiments support the effectiveness of the combination of methods they propose.

Cons:
One main concern of WvN5 is novelty. 2nYd thinks there is sufficient novelty in the proposed metric for outlier weights. I agree that there is sufficient novelty, but only if the benefits of their approach come from this metric.

Authors need to double check the statistical significance claims that they are implicitly making by bolding their results, when they computed confidence intervals on request. They cannot claim statistical significance for many of these. (eg, 33.6 ± 0.09 is not significantly worse than 33.6 ± 0.10)

Overall, this paper will be publishable once it has ablation experiments for the components of the proposed method to clarify how much contribution the novel metric provides. If these experiments show that the novel metric is beneficial, then the paper will be greatly improved. These experiments must include statistical significance tests, as currently the standard deviations presented do not suggest that the method is a robust improvement across all tasks.

**Additional Comments On Reviewer Discussion:**

Looking at the official comment by the authors that adds the LISA-D baseline, OwLore only beats all baselines on 3/8 tasks, with OwLore (Full-Rank) adding another 2 tasks. This doesn’t seem to be strong evidence for their method, especially given the fairly high standard deviations computed elsewhere on request. This might be a useful and effective method, but the evidence presented in this paper does not quite give me confidence in it.

---

### Decision · Program_Chairs · 2025-01-22

Reject